# ReViT: Rotational-equivariant Vision Transformers for Neural PDE Solvers

**Hao Wei**[1]  **Bjoern List**[1]  **Nils Thuerey**[1]

## Abstract

Physics obeys strict symmetries like rotational equivariance. However, the standard Transformer architectures widely used in physics foundation models do not enforce these constraints by construction. We introduce ReViT, a rotationally equivariant Vision Transformer framework for neural PDE solvers operating on grid-based physical fields that achieves exact equivariance for the discrete groups $C_4$ (2D) and the chiral octahedral group $O$ (3D), with bounded approximate $SO(d)$ equivariance for continuous rotations. ReViT maps scalar and vector inputs into locally invariant representations derived from physics-based canonical bases, enabling the use of standard self-attention without symmetry violations. Built on a hierarchical Swin-style backbone with a precomputed reference basis pyramid, ReViT preserves equivariance across multi-scale operations. We evaluate ReViT on a wide range of 2D and 3D PDE benchmarks, such as Magnetohydrodynamics and Turbulent Channel Flows, demonstrating significant gains over state-of-the-art baselines. ReViT exhibits strong generalization, and reduces MSE by up to 65% compared with the best-performing alternatives.

## 1. Introduction

The transformative impact of Transformer architectures has been profound in building foundation models across domains (Dosovitskiy, 2020; Liu et al., 2021). For scientific applications, considering the diverse nature of inputs, different backbones have been adapted. For unstructured data, graph-based Transformers and message-passing networks have emerged as powerful tools (Fuchs et al., 2020; Hutchinson et al., 2021). These models excel at handling irregular geometries and symmetries. In contrast, numerous scien-

tific datasets, particularly for partial differential equations (PDEs), are naturally defined on regular grids (Takamoto et al., 2022; Gupta & Brandstetter, 2022; Koehler et al., 2024; Ohana et al., 2024). Thus, vision-based Transformers, such as the Vision Transformer (ViT) and Swin Transformer, have been increasingly applied to grid-structured physics data, demonstrating strong performance in surrogate modeling (Herde et al., 2024; Holzschuh et al., 2025b; McCabe et al., 2024). In scientific applications, equivariance to symmetries is a critical property, as physical laws remain invariant under transformations like rotations (Bronstein et al., 2021; Puny et al., 2023). Rotational equivariance ensures that predictions transform consistently with input orientations. Formally, a model $f$ is equivariant to a group $G$ if $f(g \cdot x) = g \cdot f(x)$ for all $g \in G$. Rotational equivariance is a property of the governing equations themselves: rotating an entire physical system (including geometry and boundary conditions) requires the solution to rotate accordingly, as any numerical solver would reproduce. Recent work identifies enforcing such symmetries as a foundational pillar for trustworthy AI in computational fluid dynamics (Bauerheim, 2026).

These symmetries can be enforced by various approaches, ranging from soft constraints via data augmentation to hard architectural biases. Steerable and group-equivariant CNNs provide the latter through constrained kernels, offering exact equivariance for discrete groups (Cohen & Welling, 2016; Weiler & Cesa, 2019). For Graph neural networks, invariant graph networks and Tensor Field Networks (Cohen & Welling, 2016; Weiler & Cesa, 2019) provide graph equivariance. But all lack the potential of scaling up to a foundation model level. The standard self-attention inherently encodes permutation equivariance but lacks other symmetries. To overcome such drawbacks, existing equivariant Transformers predominantly focus on unstructured data, such as point clouds in SE(3)-Transformers or LieTransformers (Fuchs et al., 2020; Hutchinson et al., 2021). Existing vision-based variants are generally limited to scalar inputs for classification tasks (Romero & Cordonnier, 2020; Xu et al., 2023). To our knowledge, no Transformer architecture currently enforces rotational equivariance on grid-discretized physical data while natively supporting multi-type fields, such as scalars and vectors.

In this work, we introduce ReViT, a Transformer frame-

---

[1]Technical University of Munich. Correspondence to: Hao Wei <hao.wei@tum.de>.

*Proceedings of the 43rd International Conference on Machine Learning*, Seoul, South Korea. PMLR 306, 2026. Copyright 2026 by the author(s).

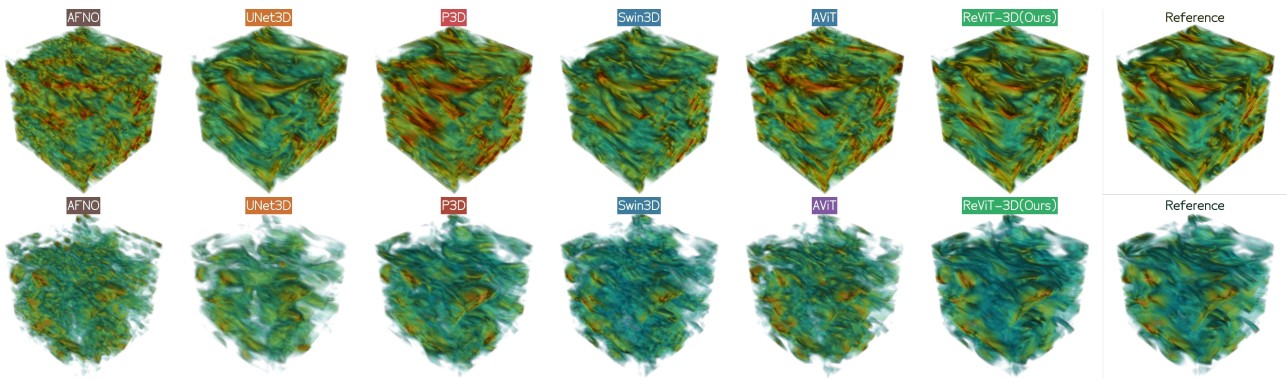

*Figure 1.* Qualitative comparison for the large-scale 3D `MHD` test case under a $\pi$ rotation around the z-axis (top: magnetic field, bottom: velocity). ReViT accurately captures fine-scale details that are smoothed out or distorted by baselines.

work that addresses this gap by extending vision backbones with hard-constrained rotational equivariance for grid-based inputs. Our contributions are summarized as follows:

1. **ReViT: Rotationally Equivariant Vision Transformers.** We introduce ReViT, the first Transformer architecture that is natively rotationally equivariant for grid-based scalar and vector fields. Unlike prior equivariant Transformers targeting unstructured particles or graphs, ReViT retains the inductive bias of regular grids, making it particularly suited for scientific data such as fluid flows and physical fields.

2. **Architecture-Agnostic Invariant Embedding.** Drawing from the strain rate tensor in fluid mechanics, we propose to compute a rotation-invariant local coordinate system per patch, yielding invariant tokens that enable the use of standard, optimized multi-head self-attention without compromising physical symmetry.

3. **Hierarchical Reference Basis Pyramid.** We present a Swin-inspired hierarchical backbone with a precomputed Reference Basis Pyramid for equivariant multi-scale operations. This ensures strict equivariance across resolutions, remains agnostic to underlying ViT-style architectures.

4. **Extensive Validation on Diverse Physics Tasks.** We comprehensively evaluate ReViT across a broad range of benchmarks, demonstrating strong generalization and equivariance: rotated MNIST (scalar classification), 2D advection (scalar rollout prediction), and other vector rollout predictions like 2D Kolmogorov flow, 3D turbulent channel flow, and 3D magnetohydrodynamics.

## 2. Related Work

**Equivariance by canonicalization.** Several works achieve equivariance by transforming data into a canonical reference frame, including a global canonical orientation (Zhao et al., 2022; 2020; Baker et al., 2024; Li et al., 2021; 2025b), and a local canonicalization (Wang & Zhang, 2022; Luo et al., 2022; Du et al., 2023; Lippmann et al., 2025; List et al., 2025). These approaches rely on graph-based message passing, resulting in quadratic complexity and high memory demands that overlook the regular grid structure.

**Group CNNs.** By lifting standard convolutional operations to symmetry groups, CNNs have been extended to broader symmetry groups (Cohen & Welling, 2016). This paradigm has been successfully generalized to planar rotations (Dieleman et al., 2016; Marcos et al., 2017; Worrall et al., 2017; Graham et al., 2020), spherical rotations (Cohen et al., 2018; Sosnovik et al.; Romero et al., 2020b), groups in Fourier domain (Helwig et al., 2023), and more general symmetry groups (Kondor & Trivedi, 2018; Tai et al., 2019; Venkataraman et al., 2020; Bekkers, 2020; Chen et al., 2021; Finzi et al., 2020; Li et al., 2024; 2025a). However, discrete kernels tie weights to fixed positions, leading to parameters scaling significantly with resolution (Romero & Cordonnier, 2020; Xu et al., 2023).

**Equivariant transformers.** To build equivariant transformers, self-attention has been lifted into group space (Romero & Hoogendoorn, 2019; Diaconu & Worrall, 2019). However, lifting attention significantly exacerbates the parameter and memory bottlenecks, often limiting these models to small-scale classification tasks (Romero et al., 2020a; Xu et al., 2023). While other equivariant Transformers (Liao & Smidt; Hutchinson et al., 2021; Fuchs et al., 2020; Spinner et al., 2025) excel on sparse point clouds, their quadratic complexity relative to the number of input points makes them computationally intractable for dense 3D grid data.

## 3. Theoretical Analysis: Why ViTs fail

This section establishes the theoretical foundations of ReViT. We introduce the fundamental concepts of rotational equivariance and rigorously identify the structural barriers that prevent standard Vision Transformers from processing vector-valued physical fields equivariantly.

**Group Actions on Fields.** Let $\Omega \subset \mathbb{R}^d$ be a spatial domain ($d \in \{2, 3\}$). We consider a physical field $\mathbf{f} : \Omega \to \mathcal{V}$, where $\mathcal{V}$ is the feature space (e.g., $\mathbb{R}$ for scalars, $\mathbb{R}^d$ for

velocity vectors). In practice, this is represented as a discretized tensor on a grid. The symmetries of the domain are described by the Special Orthogonal group $G = SO(d)$. Each element $g \in G$ corresponds to an orthogonal rotation matrix $\mathbf{R}(g) \in \mathbb{R}^{d \times d}$. Crucially, the group action of rotation depends on the physical nature of the field. Let $L_g$ denote the induced action of $g$ on a function. For a *scalar field* $\phi : \Omega \to \mathbb{R}$, the group acts only on the spatial coordinate:

$$[L_g \phi](x) = \phi(\mathbf{R}(g)^{-1} x). \tag{1}$$

In contrast, for a *vector field* $\mathbf{u} : \Omega \to \mathbb{R}^d$, the group acts on both the domain and the value:

$$[L_g \mathbf{u}](x) = \mathbf{R}(g)\mathbf{u}(\mathbf{R}(g)^{-1} x). \tag{2}$$

A neural network $\Phi$ is *equivariant* if it commutes with this action: $\Phi(L_g \mathbf{u}) = L_g \Phi(\mathbf{u})$.

**Permutation Equivariance of SA.** Let $\mathbf{X} \in \mathbb{R}^{N \times C}$ denote a tokenized representation of the field, where each token corresponds to a spatial location $x_i \in \Omega$. The SA operator is defined as:

$$\text{SA}(\mathbf{X}) = \text{softmax}\left( \frac{\mathbf{X}\mathbf{W}_Q (\mathbf{X}\mathbf{W}_K)^T}{\sqrt{d_k}} \right) \mathbf{X}\mathbf{W}_V, \tag{3}$$

where $\mathbf{W}_Q, \mathbf{W}_K, \mathbf{W}_V$ are learned linear projections acting on the feature dimension $C$. SA is inherently *permutation equivariant*. If a rotation $g$ induces a permutation $\boldsymbol{\pi}_g \in S_N$ on the spatial tokens, it holds that: $\text{SA}(\boldsymbol{\pi}_g \mathbf{X}) = \boldsymbol{\pi}_g \text{SA}(\mathbf{X})$.

This property is sufficient for fields where the features within tokens remain invariant, such as scalar fields (Romero & Cordonnier, 2020), but is insufficient for vector fields, as detailed below.

### 3.1. Challenges to Equivariance in ViTs

We identify 3 distinct mechanisms by which standard ViTs violate rotational equivariance when applied to vector fields.

**C1. The Tokenization Barrier.** Before SA is applied, ViTs partition the domain into patches. Let $P_i \subset \Omega$ be a local patch of size $K^d$ ($d \in \{2, 3\}$). The standard patch embedding flattens the vector field within $P_i$ into a vector $\mathbf{v} \in \mathbb{R}^{K^d \cdot d}$ and applies a learnable linear map $\mathbf{E}$ to obtain $\mathbf{z}_i = \mathbf{E}(\mathbf{v})$. Crucially, this flattening operation entangles the semantic content of the patch with the specific grid ordering of its pixels. Under a rotation $g$, the pixels within $P_i$ undergo a spatial permutation $\pi_g$, changing the flattened input to $\pi_g \mathbf{v}$. To leverage the permutation equivariance of the SA, the input tokens themselves must be invariant to these local transformations. That is, we require $\mathbf{z}_i' = \mathbf{E}(\pi_g \mathbf{v}) = \mathbf{z}_i$. But, standard linear embeddings fail to satisfy this condition, as $\mathbf{E}(\pi_g \mathbf{v}) \neq \mathbf{E}(\mathbf{v})$ in the general case. Consequently, a rotated pattern is projected into a disjoint region of the latent

space, preventing the model from translating permutation equivariance into rotation equivariance.

**C2. Loss of Spatial Equivariance.** While the SA operator is permutation equivariant, it is blind to geometry. To resolve spatial ambiguity, Positional Encodings (PEs) are introduced, but standard implementations fundamentally break rotational equivariance:

**Absolute PEs break equivariance:** Standard ViTs use fixed absolute encodings $\mathbf{P}(x)$. For a rotation $g$, the position $x$ transforms to $\mathbf{R}(g)x$, but the stored encoding remains $\mathbf{P}(x)$. Thus, $\mathbf{P}(\mathbf{R}(g)x) \neq \boldsymbol{\pi}_g \mathbf{P}(x)$, preventing the model from recognizing the rotated object.

**Relative PEs lose directionality:** Relative PEs $\psi(x_i, x_j)$ depend on the displacement vector $\boldsymbol{\delta}_{ij} = x_j - x_i$. It restores translation equivariance (as $(x_i + t) - (x_j + t) = x_i - x_j$), but struggles with rotational equivariance. As derived in (Romero & Cordonnier, 2020), ensuring rotational equivariance requires the PEs to be invariant to the group action, $\psi(\mathbf{R}(g)x_i, \mathbf{R}(g)x_j) = \psi(x_i, x_j), \forall g \in G$. An *isotropic* function, like $\|x_i - x_j\|$, fits this constraint. But it discards all directional orientation information. Thus, PEs break rotational equivariance or eliminate orientation information.

**C3. The Representational Mismatch.** The most fundamental challenge arises from how rotations act on feature values. For a scalar field, a rotation $g$ acts solely on coordinates, resulting in a rearrangement of pixels, leaving the value unchanged. On a discrete grid, this induces a permutation $\boldsymbol{\pi}_g$. Since SA commutes with $\boldsymbol{\pi}_g$, the operation is theoretically rotation equivariant. Thus, for scalar fields, ViTs can achieve rotational equivariance once spatial indexing is handled appropriately. GSA and GE-ViT (Romero & Cordonnier, 2020; Xu et al., 2023) achieved rotational equivariance by leveraging the permutation equivariance of SA with *lifting* the positional encoding to the group $G$.

In contrast, for a vector field, even if the spatial positioning issues were resolved, a fundamental structural failure remains. As shown in Equation (2), rotations act on both coordinates and values, indicating that for the network to be equivariant to vector inputs, the linear projections must commute with the geometric rotation:

$$\mathbf{W}(\mathbf{R}(g)\mathbf{u}) = \mathbf{R}(g)(\mathbf{W}\mathbf{u}). \tag{4}$$

By Schur's Lemma (Serre et al., 1977), any linear map commuting with an irreducible representation, like rotations of vectors, must be a scalar multiple of the identity ($\mathbf{W} = \lambda \mathbf{I}$). This creates a fundamental conflict: a standard linear layer $\mathbf{W}$ cannot mix vector components without breaking equivariance. As ViT projection matrices are unconstrained and dense, which violates Equation (4) and breaks equivariance.

## 4. Methodology of ReViT

In this section, we introduce the ReViT architecture. By adapting local canonicalization to ViTs, we decouple basis transformations from feature learning, effectively solving challenges C1 to C3. As illustrated in Figure 2, the model flow alternates between two distinct states: **invariant processing** (depicted in blue), where features are invariant within local canonical frames, and **global transitions** (depicted in orange), where the physical basis is transformed. The overall architecture consists of 3 stages: (1) Local Canonicalization, (2) Invariant Transformer Processing, and (3) Equivariant Decoding.

### 4.1. Local Canonicalization for Representational Mismatch

To resolve *Representational Mismatches* (C3),we abandon global Cartesian coordinates in favor of local, physics-based reference frames. Let $P_i$ be a local patch of the input vector field **u**. Our core mechanism is the computation of a *Local Canonical Basis* $\mathbf{B}_i \in SO(d)$, which is derived deterministically from the field values within the patch. Given such a basis, we project the physical vectors $\mathbf{u}_k$ (where $k \in P_i$) into the local frame: $\mathbf{u}_{i,k}^{\text{local}} = \mathbf{B}_i^T \mathbf{u}_k$.

**Proposition 4.1** (Invariance of Local Projection). *Let $\phi(\mathbf{B}_i^T \mathbf{U}_i)$ be the patch embedding where $\mathbf{B}_i \in \text{SO}(d)$ is the local canonical basis derived from the input field. Then for all $R \in \text{SO}(d)$: $\phi(\mathbf{B}_i'^T(R\mathbf{U}_i)) = \phi(\mathbf{B}_i^T \mathbf{U}_i)$, where $\mathbf{B}_i' = R\mathbf{B}_i$.*

**Proof of Invariance:** Since $\mathbf{B}_i$ is based on the input field **u**, it rotates accordingly: $\mathbf{u}_i' \mapsto \mathbf{R}(g)\mathbf{u}_i \implies \mathbf{B}_i' \mapsto \mathbf{R}(g)\mathbf{B}_i$. Under a global rotation $g \in SO(d)$, $\mathbf{u}_{i,k}^{\text{local}}$ become invariant to global rotations:

$$(\mathbf{R}(g)\mathbf{B}_i)^T(\mathbf{R}(g)\mathbf{u}_i^{\text{global}}) = \mathbf{B}_i^T \mathbf{R}(g)^T \mathbf{R}(g)\mathbf{u}_i^{\text{global}}$$
$$= \mathbf{B}_i^T \mathbf{u}_i^{\text{global}} = \mathbf{u}_{i,k}^{\text{local}} \quad (5)$$

After re-basing, vector components become *invariant scalars*. Consequently, standard linear projections **W** can be applied without violating equivariance constraints, resolving the representational mismatch.

Unlike previous methods relying on the computationally heavy eigen-decomposition of the strain tensor (List et al., 2025), we derive the basis directly from macroscopic directional fields. In 3D, we utilize the mean velocity $\bar{\mathbf{u}}_i$ and mean vorticity $\bar{\boldsymbol{\omega}}_i$. To ensure strict orthogonality and computational efficiency, we employ a stabilized analytical orthogonalization based on sequential cross-products. It deterministically constructs the basis vectors while explicitly handling singularities, such as collinear fields, and is applicable to general vector fields. Additionally, we also detail a robust adaptation of the strain-based approach for grid-based Transformer architectures in Appendix C

We emphasize that the canonical basis can be constructed

from *any* equivariantly transforming vector quantity, not exclusively fluid velocity. Our experiments already demonstrate this generality: RotMNIST uses the spatial gradient $\nabla\phi$ of the scalar image field (Appendix C.3), MHD uses the magnetic field rather than velocity. The only requirement is non-degeneracy ($\|\bar{\mathbf{u}}_i\| > \epsilon$); degenerate cases are handled by deterministic fallback procedures detailed in Appendix C. A further discussion on the generality of ReViT, including extensions to higher-order tensor fields, is provided in Appendix J.

### 4.2. Invariant Embeddings for Tokenization Barrier

Instead of flattening the patch into a fixed-order vector, which breaks under rotation-induced permutations (C1), we treat the projected local vectors within patch $P_i$ as a set of invariant feature vectors $\mathcal{X}_i = \{\mathbf{h}_{i,k} \mid k \in P_i\}$, where $\mathbf{h}_{i,k} = \text{MLP}(\mathbf{u}_{i,k}^{\text{local}})$. To aggregate this set into a fixed-size representation while preserving permutation invariance, we adapt a Set-Transformer (Lee et al., 2019). Firstly, a learnable latent token $\mathbf{h}_{\text{query}} \in \mathbb{R}^D$ is concatenated with the set of invariant features to form the input sequence $\mathbf{H} = [\mathbf{h}_{\text{query}}, \mathbf{h}_1, \ldots, \mathbf{h}_{K^d}]$. We then apply a standard SA layer $\mathbf{H}' = \text{SA}(\mathbf{H})$, and extract the query token, $\mathbf{z}_i = \mathbf{H}'[0]$. Consequently, the operation is strictly permutation invariant:

$$\text{Agg}(\pi(\mathcal{X}_i)) = \text{Agg}(\mathcal{X}_i). \quad (6)$$

This ensures that even as the locations permute under rotation, the resulting patch token remains numerically identical, ensuring a rotation-invariant input to the main transformer.

### 4.3. Relative Rebasing for Spatial Equivariance

To address the *Loss of Spatial Equivariance* (C2), we fundamentally alter how position embeddings (PEs) are injected into the attention mechanism. Standard relative PEs lost directional information. In ReViT, since the relative PEs depend on the displacement vector $\boldsymbol{\delta}_{ij} = x_j - x_i$, we define the *rebased relative position* by projecting $\boldsymbol{\delta}_{ij}$ into the query token's basis $\mathbf{B}_i$: $\mathbf{p}_{ij \to i} = \mathbf{B}_i^T \boldsymbol{\delta}_{ij} = \mathbf{B}_i^T(x_j - x_i)$.

By construction, $\mathbf{p}_{ij \to i}$ is invariant to global rotations while remaining anisotropic in the query-local frame, thereby preserving directional information relative to the local flow. This vector is mapped to a scalar positional bias via an MLP, $\mathbf{P}_{ij} = \text{MLP}_{\text{pos}}(\mathbf{p}_{ij \to i})$. Collecting $\mathbf{P}_{ij}$ for all token pairs yields an attention bias matrix $\mathbf{P} \in \mathbb{R}^{N \times N}$, which is added to the dot-product attention logits:

$$\text{SA}(\mathbf{X}) = \text{softmax}\left(\frac{\mathbf{X}\mathbf{W}_Q(\mathbf{X}\mathbf{W}_K)^T}{\sqrt{d_k}} + \mathbf{P}\right)\mathbf{X}\mathbf{W}_V. \quad (7)$$

As the PEs enter the attention only through a bias term, the

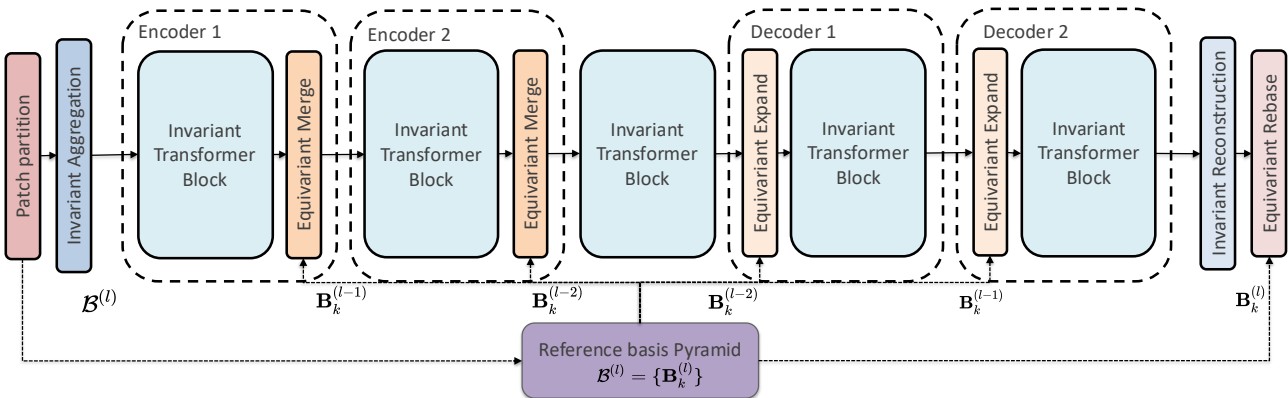

*Figure 2.* Overview of the ReViT architecture. The hierarchical encoder-decoder alternates between **invariant processing** (blue) and **global transitions** (orange). The *Reference Basis Pyramid* (purple) acts as the bridge between these states, supplying local bases $\mathcal{B}^{(l)}$ to mediate resolution changes (*Merge*, *Expand*) and the final *Equivariant Rebase*.

resulting SA operator preserves permutation equivariance while remaining equivariant to global rotations.

### 4.4. From Invariance to Equivariance: Equivariant Decoding

The transformer outputs a sequence of processed invariant tokens $\mathbf{h}_i^{(L)}$, which effectively encapsulate the conditional physics of each patch. To recover the physical vector field $\mathbf{u}'$ in the global frame, we must reintroduce the spatial structure and orientation. We employ a *local query decoder* that reverses the encoder's aggregation process. Following the continuous reconstruction paradigm of CViT (Wang et al., 2024), we define a fixed canonical grid of coordinates $\mathcal{G} = \{\boldsymbol{\xi}_m \in [-1,1]^d\}_{m=1}^{P^d}$. These coordinates are mapped to high-dimensional embeddings via Fourier features to serve as *spatial queries* $\mathbf{Q}_{\text{grid}}$. We then apply a cross-attention mechanism to reconstruct the dense spatial details from the compressed semantic information (Wang et al., 2024; Koupaï et al., 2025). $\mathbf{z}_i^{\text{local}} = \text{CrossAttn}\left(\mathbf{Q}_{\text{grid}}, \mathbf{h}_i^{(L)}\right)$.

To achieve global equivariance, we lift the predictions back to the global coordinate system using the preserved local basis $\mathbf{B}_i$: $\mathbf{u}_i' = \mathbf{B}_i \cdot \mathbf{z}_i^{\text{local}}$. Because the local predictions $\mathbf{z}^{\text{local}}$ are invariant to the input rotation, and $\mathbf{B}_i$ rotates concurrently with the input, the final output is strictly equivariant.

### 4.5. Hierarchical Equivariance via Reference Basis Pyramids

To capture features at varying resolutions, we extend ReViT into a hierarchical backbone inspired by the Swin Transformer (Liu et al., 2021). But standard patch merging (e.g., pooling) is mathematically invalid for our tokens, as averaging vectors defined in disparate local bases breaks physical consistency. To resolve this, we introduce a *reference basis pyramid*, a pyramid of local canonical bases $\mathcal{B}^{(l)} = \{\mathbf{B}_k^{(l)}\}$,

pre-computed at each resolution $l$ from the input field $\mathbf{u}$. Then we employ a "globalize-resample-localize" procedure: before changing resolution, invariant features are projected back to a global coordinate frame. Once aligned globally, they undergo valid spatial operations like pooling or interpolation before being immediately re-projected into the target resolution's local bases. This ensures that features remain rotation-invariant across scales, allowing ReViT to function as a hierarchical backbone with a strict equivariance chain. Specifically, for **patch merging** (downsampling), we apply average pooling with kernel size $2^d$ and stride 2 in the global frame, followed by LayerNorm and a linear layer for channel expansion. For **patch expansion** (upsampling), we use trilinear interpolation to upsample features by a factor of 2, fused with encoder skip connections via concatenation, followed by LayerNorm and a linear layer for channel reduction.

The hierarchical design also mitigates potential boundary effects under arbitrary rotations. When pixels shift across patch boundaries due to rotation, the shifted window attention mechanism propagates information across adjacent patches, averaging out boundary inconsistencies. Hierarchical merging further absorbs these effects: boundary inconsistencies at one resolution are subsumed into shared patches at the next coarser level. The reference bases at each level are pre-computed from the input and co-rotate with it, ensuring self-consistency. A discussion of why local bases are preferred over a single global basis is provided in Appendix I.

In summary, ReViT resolves representational mismatches by projecting vectors into local canonical bases, yielding invariant embeddings compatible with standard SA via a permutation-invariant aggregator. Note that the position of pixels within a patch is not used for computing the invariant embedding, consistent with standard ViT/Swin practice;

empirically, adding intra-patch positional encoding does not improve and can slightly hurt performance (see Appendix J.3). We preserve spatial structure using rebased relative PEs that maintain local anisotropy. The architecture concludes with an equivariant decoder that lifts predictions back to global, while a reference basis pyramid mediates resolution changes to ensure hierarchical physical consistency. An ablation study of the architectural design is detailed in Appendix B.

**Analysis of Equivariance.** Strictly speaking, our ReViT achieves exact chiral octahedral group $O$ equivariance and approximate $SO(3)$ equivariance. The gap stems from unavoidable grid-based constraints: resampling introduces an interpolation bias, and discretization artifacts arise from fixed patch/window boundaries. Unlike $\frac{\pi}{2}$, arbitrary rotations introduce interpolation bias and force features to cross these boundaries irregularly, breaking grid symmetry. Thus, we focus on the effectiveness of guaranteeing equivariance for the discrete chiral octahedral group $O$ as a robust proxy for the continuous group $SO(3)$. Meanwhile, we investigate whether augmentation helps ReViT dampen these discretization artifacts, thereby enhancing robustness without claiming strict mathematical validity. We formalize this gap in Appendix G by showing that the equivariance defect is bounded by $\|f(\mathcal{I}_h(R\mathbf{u})) - Rf(\mathbf{u})\| \leq L_f \cdot \epsilon_{\text{resamp}}$, where $\epsilon_{\text{resamp}}$ vanishes for $R \in O$ and decreases as $O(h^p)$ with grid spacing $h$.

We empirically validate the rotational equivariance over the 24 elements of the chiral octahedral group $O$ on 3D data. The equivariance defect is evaluated independently for each $g \in O$ and averaged over all test samples, without any optimization over group elements. Using the error metric $\mathcal{E}(\mathbf{U}; g) = \|f(g \cdot \mathbf{U}) - g \cdot f(\mathbf{U})\|$ and $R^2(g) = 1 - \frac{\sum_n |f(g \cdot \mathbf{u}_n) - g \cdot f(\mathbf{u}_n)|^2}{\sum_n |g \cdot f(\mathbf{u}_n) - \overline{g \cdot f(\mathbf{u}_n)}|^2}$ as a normalized similarity measure, ReViT-3D achieves perfect equivariance ($R^2 = 1.0$) across all 24 configurations (Figure 3). Conversely, baselines exhibit near-zero correlations for non-identity ($O_0$) rotations, indicating a fundamental lack of equivariance in standard backbones. Note that $O$ is the maximal finite rotation subgroup of $SO(3)$ compatible with a cubic grid (Balla et al., 2025); further details on the metric are in Appendix H.

## 5. Results

To comprehensively validate the effectiveness of ReViT, we evaluate it across a diverse set of datasets and compare it against established baselines. To demonstrate the flexibility of our method, which is designed to integrate seamlessly with existing successful architectures, we implement ReViT using two distinct backbones. The first is the standard Vision Transformer (Dosovitskiy, 2020), deployed for the rotational classification task on RotMNIST. The second is

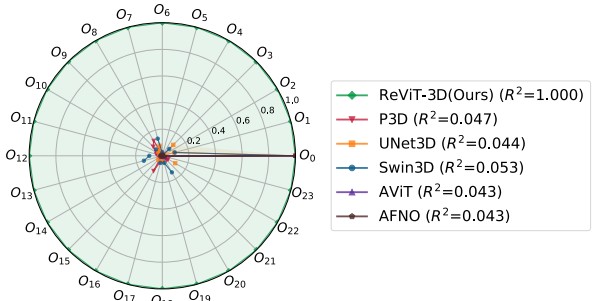

*Figure 3.* Rotational equivariance evaluation over the chiral octahedral group $O$. The radial axis represents the $R^2$ score between the rotated model output and the model output on rotated inputs. While baselines fail to generalize to rotations ($R^2 \approx 0$), ReViT-3D maintains perfect symmetry ($R^2 = 1.0$).

a Swin-based architecture (Liu et al., 2021) used for scientific foundation models in the form of PDE-Transformer in 2D (Holzschuh et al., 2025b) and 3D (Holzschuh et al., 2025a), which we apply to PDE-based prediction tasks.

### 5.1. Experimental Configurations

**RotMNIST.** We begin with RotMNIST, a canonical benchmark for evaluating rotation equivariance. It consists of 62k gray-scale ($28 \times 28$) handwritten digits, uniformly rotated. Herein, we benchmark ReViT against two state-of-the-art equivariant transformer architectures: GSA-Nets (Romero & Cordonnier, 2020) and GE-ViT (Xu et al., 2023).

**2D Advection.** The second dataset, denoted as Adv, is derived from the AdvBox benchmark (Lino et al., 2022a;b; List et al., 2025) to allow for comparisons with equivariant GNNs. The training set comprises 1500 simulations, with 20% used for validation, while the test set consists of 200 simulations. The original unstructured data was resampled onto a regular grid for the training of Transformers as detailed in F.2. The physical setup involves a passive scalar quantity $\phi$ advected by a constant velocity field $\mathbf{u}$ within a periodic domain. Here, $\phi$ represents the sole dynamic quantity. We compare ReViT against equivariant networks, including a data-augmented, regular GNN (denoted by GNN-Aug), ReGNN (List et al., 2025), REMUS (Lino et al., 2022b), SE3GN (Brandstetter et al., 2021), and an equivariant transformer-based model, LGATrans (Spinner et al., 2025). Moreover, we include a regular, non-equivariant PDE-Transformer (Holzschuh et al., 2025b) due to its architectural similarities with ReViT. For this comparison, the graph-based models are evaluated on graph-structured meshes, in accordance to their training data (Lino et al., 2022b), while the transformers operate on regular mesh.

**2D Kolmogorov Flow.** The third dataset, denoted as KF, represents a 2D Kolmogorov Flow scenario sourced from the APEBench benchmark (Koehler et al., 2024). The task involves predicting the time evolution of a velocity field under spatially periodic sinusoidal forcing. The resolution

*Table 1.* Comparison of accuracy and computational costs. Time costs are reported in milliseconds per batch. Memory represents the total VRAM usage by 1 GPU with a batch size of 32.

| MODEL | ACC (%) | TR. (MS) | INF. (MS) | MEM (GB) |
|---|---|---|---|---|
| GSA-NETS($R_4$) | 97.46 | 298.8±0.9 | 110.0±0.1 | 5.27 |
| GSA-NETS($R_8$) | 97.90 | 144.2±2.1 | 65.2±0.2 | 29.9 |
| GSA-NETS($R_{12}$) | 97.97 | 272.6±0.6 | 118.7±0.5 | 95.5 |
| GE-VIT($R_{12}$) | 98.01 | 281.0±0.7 | 118.9±0.3 | 95.5 |
| REVIT(OURS) | **98.26** | **67.7±0.2** | **31.0±0.7** | **1.81** |

of the velocity field is $160^2$, and the dynamics are governed by the incompressible Navier–Stokes equations with periodic boundary conditions as detailed in Appendix F. The periodic nature of the domain allows for arbitrary rotations without padding artifacts, makes KF an ideal testbed for examining the performance of equivariant models.

**3D Magnetohydrodynamics Compressible Turbulence.** The fourth dataset, denoted as MHD, focuses on 3D compressible magnetohydrodynamic turbulence sourced from The Well dataset (Ohana et al., 2024; Burkhart et al., 2020), with a spatial resolution of $64^3$. For MHD, the input is more complex, consisting of a scalar density field and two vector fields: the velocity and the magnetic field. Since there are two vector fields, to demonstrate the robustness of local canonicalization, we use the magnetic field rather than the velocity to rebase all the vectors. The full governing equations are provided in Appendix F. For our experiments, we set the Sonic Mach number to $M_s = 0.5$ and the Alfvénic Mach number to $M_A = 0.7$. Here, we evaluate ReViT against a comprehensive suite of benchmarks, including P3D (Holzschuh et al., 2025a), AViT (McCabe et al., 2024), Adaptive FNO (AFNO, (Guibas et al., 2021)), Swin3D, our own implementation of SwinV2 (Liu et al., 2022), and a 3D variant of UNet (Ronneberger et al., 2015).

**3D Turbulent Channel Flow.** The final dataset, denoted as TCF, represents a periodic channel with no-slip boundaries at the walls ($y$-axis) driven by dynamic forcing to maintain energy levels against wall friction. This dataset is a widely used benchmark for PDE prediction (Holzschuh et al., 2025a; Franz et al., 2026; Wei et al., 2025; Balla et al., 2025). We consider a Reynolds number 400, with 200 snapshots captured at $\Delta t = 0.1$ on a $96^3$ spatially adaptive grid. The benchmarks kept the same with MHD dataset.

**5.2. Classification Performance**

We begin by evaluating ReViT against equivariant Vision Transformers on the canonical benchmark rotMNIST. Herein, GSA-Nets, utilizing group discretizations of size 4, 8, and 12 (denoted as $R_4$, $R_8$, $R_{12}$), and GE-ViT with $R_{12}$.

As shown in Table 1, ReViT achieves SOTA accuracy (98.26%) while delivering a ∼4× speedup and a ∼53×

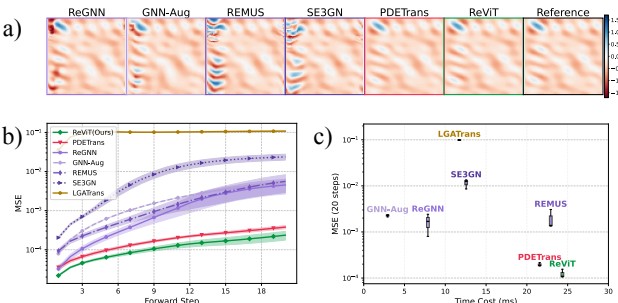

*Figure 4.* Evaluations for Adv: a) Qualitative at rollout step 20. ReViT closely matches the reference by capturing details more accurately than baselines, which show artifacts near the boundary. b) Temporal stability over 20 autoregressive steps. ReViT exhibits the highest stability compared to baselines. c) Computational efficiency vs prediction accuracy tradeoff: ReViT (green) achieves the lowest error with competitive latency.

memory reduction (1.81 GB vs. 95.5 GB) compared to the $R_{12}$ baselines. The baselines' inefficiency stems from the "lifting" operation ($\mathbb{R}^d \to \mathbb{R}^d \rtimes \mathcal{H}$), which expands feature maps by the stabilizer size $|\mathcal{H}|$. This inflates self-attention complexity to $\mathcal{O}(N^2|\mathcal{H}|^2)$, a constraint also present in GE-ViT (Xu et al., 2023). These results demonstrate that the explicit lifting procedure incurs unsustainable computational costs, rendering these architectures infeasible for larger-scale data. Thus, we exclude lifted architectures from the subsequent high-dimensional scientific experiments.

**5.3. PDE-focused Architectural Comparisons**

As rotationally equivariant graph-based architectures are popular for PDE-based solvers, it is interesting to analyze advantages and disadvantages of graph and transformer based architectures in the context of PDE inference. For this, we evaluate ReViT's performance against state-of-the-art baselines across prediction accuracy, computational efficiency, and long-term stability based on the Adv test case.

The quantitative evaluation of long-term rollout stability is presented in Figure 4(b). Our ReViT achieves the lowest MSE ($\approx 10^{-4}$) among all compared methods, while the baseline GNN methods (ReGNN, REMUS, SE3GN) exhibit a stronger error accumulation and high variance across different random seeds. LGATrans, as a recently proposed general approach for equivarance, likewise yields suboptimal accuracy. This is caused by its regular transformer backbone, which, despite being equivariant, lacks the inductive biases of ViT architectures for spatio-temporal tasks.

Crucially, ReViT consistently outperforms its closest architectural competitor, the non-equivariant PDETrans (red curve), which highlights the specific contribution of the proposed equivariant mechanisms introduced in our paper. The inference latency against the 20-step cumulative MSE is

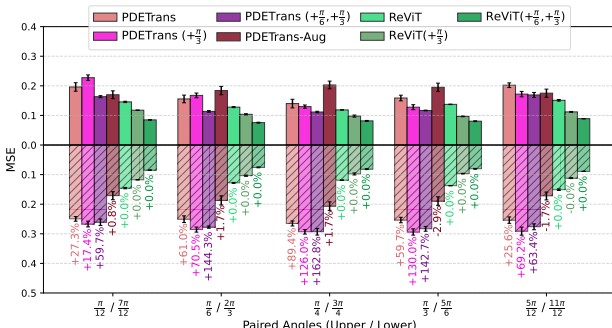

*Figure 5.* The bars represent the MSE at paired orthogonal angles: at the top top for $\theta$, and bottom for $\theta + \frac{\pi}{2}$. Both bars should be identical for an equivariant model. The text annotations indicate the relative error percentage between the pair. ReViT achieves 0.0% relative error (perfect equivariance), while PDETrans (purple/pink) shows high variance(up to +162.8%) on unseen angles.

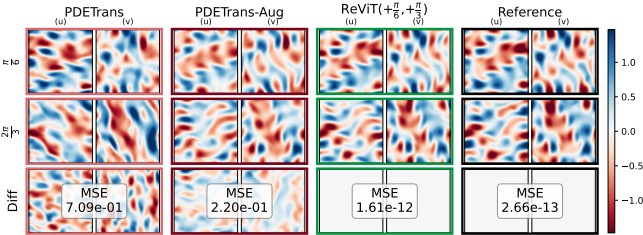

*Figure 6.* Predicted velocity field $(u, v)$ of an orthogonal angular pair after 20 steps rollout, and an error map shows the difference between the rotation of the $\frac{\pi}{6}$ and $\frac{2\pi}{3}$. ReViT achieves perfect equivariance, but PDETrans variants show strong artifacts.

shown in Figure 4(c). As shown in Figure 4(a), all GNN methods exhibit significant artifacts near the left boundary which periodically wraps to the right, and contains a finer cells in the underlying graph. While PDETrans captures the general structure, ReViT yields sharper boundaries and clearer separation between wave peaks, further validating the efficacy of embedding geometric symmetries directly into the transformer backbone. The computational cost shown in Figure 4(b) shows that the lightweight GNN architectures as proposed for this test scenario (List et al., 2025) offer lower inference latency, but they suffer from error rates that are an order of magnitude higher (MSE $\approx 10^{-3}$). The computational overhead of ReViT is comparable to the non-equivariant PDETrans with a 11.6% increase, yet it delivers superior accuracy by a reduction in MSE around 37.2%, confirming that the proposed method improves performance without introducing prohibitive computational costs.

## 5.4. Robustness on Arbitrary Angles via Data Augmentation

To evaluate model performance under continuous rotation and the effect of data augmentation, we analyze prediction accuracy over 20 rollout steps across angular intervals of $\frac{\pi}{12}$

within the range $(0, \pi)$. Our evaluation focuses on orthogonal angular pairs ($\theta$ and $\theta + \frac{\pi}{2}$). A grid-based rotation by $\frac{\pi}{2}$ has no discretization artifacts, and hence an equivarant model should yield identical performance. We use the relative difference in MSE between these orthogonal pairs to quantify equivariance errors. To study the impact of data augmentation, we compare against two baseline strategies: random angular augmentation (PDETrans-Aug) and specific discrete training angles (PDETrans ($+\theta$)).

As illustrated in Figure 5, ReViT demonstrates perfect equivariance for all orthogonal angular pairs (top and bottom bars for rotated input match). The relative error for all ReViT configurations is exactly **+0.0%**, regardless of the input angle $\theta$. This confirms that ReViT successfully preserves $C_4$ symmetry, rendering the model mathematically equivariant to discrete $\frac{\pi}{2}$ rotations. This is qualitatively verified in Figure 6, where the difference map between the orthogonal pair for ReViT only shows floating point noise. Furthermore, ReViT bridges the gap from discrete $C_4$ symmetry to continuous $SO(2)$ generalization with minimal data augmentation. While artifacts prevent strict equivariance for non-orthogonal rotations, ReViT maintains a consistent performance. From Figure 5, even the base ReViT remains stable across unseen angles, and additional training data further reduces errors.

Relying solely on data-driven invariance, the data-augmented PDETrans baselines lead to larger errors and unstable generalization. The randomly augmented model (*PDETrans-Aug* in brown) shows a relatively stable performance, but exhibits a $130.8\%$ larger MSE compared to the best ReViT variant. It additionally produces blurry predictions, as shown in Figure 6. With specific discrete angles, there is severe overfitting. E.g., PDETrans ($\frac{\pi}{6}, \frac{\pi}{3}$) in Figure 5 exhibits an error spike of +162.8% when evaluated on orthogonal pairs. This stark contrast highlights that without inductive biases for equivariance, standard transformers merely memorize specific orientations via data augmentation, lacking underlying physical symmetries.

## 5.5. Generalization via Equivariance in 3D

We evaluate ReViT and competing benchmarks on two 3D tasks in terms of their generalizing capabilities on the chiral octahedral group $O$. The statistical isotropy of turbulence provides strong *implicit rotational augmentation* (Balla et al., 2025), theoretically allowing standard networks to learn rotational symmetries from data. However, the effectiveness of this implicit learning is limited by the physical constraints of the flow. While the mean magnetic field introduces global anisotropy in MHD, the periodic boundary conditions imply spatial homogeneity, and the turbulent cascade provides partial implicit augmentation. TCF represents a symmetry-starved regime: the wall-bounded geometry and

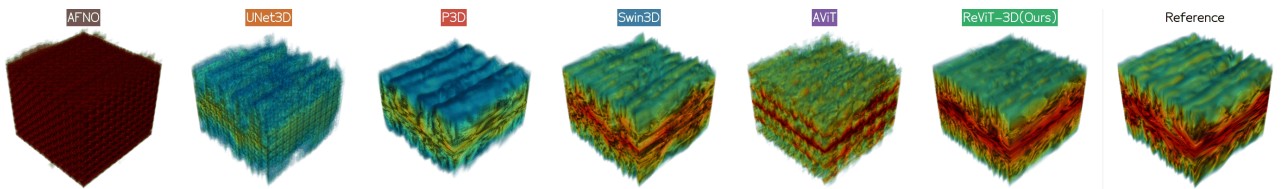

*Figure 7.* Qualitative comparison for `TCF` under a $\pi$ rotation around z. ReViT closely matches the reference solution, faithfully capturing near-wall turbulent streaks. Baselines introduce significant artifacts and fail to maintain flow continuity near the boundaries.

dominant streamwise flow induce severe spatial anisotropy, effectively eliminating implicit rotational augmentation. By comparing ReViT against benchmarks on those two tasks, we aim to show that (1) hard-coding equivariance yields superior performance; and (2) in highly anisotropic cases, explicit equivariance is not just beneficial, but strictly necessary for correct physical modeling.

**3D MHD.** ReViT achieves a decisive improvement over all baseline architectures, recording the lowest MSE (0.82 $\times 10^{-2}$) and the highest $R^2$ (0.98), as detailed in Table 2. It outperforms the strongest baseline, AViT, by approximately 63% in MSE and surpasses standard backbones like UNet3D and Swin3D by a significant margin. Qualitatively, from the visual comparison in Figure 1, ReViT preserves the sharp, high-frequency structures of the magnetic field and velocity eddies, remaining virtually indistinguishable from the reference simulation. Among the baselines, AViT performs relatively well; its Axial Attention mechanism aggregates information along axes (channels), which can arguably better capture relevant features under rotation. However, AViT still lacks the geometric structural equivariance, as detailed in Section 4, resulting in it still introducing subtle aliasing and distortions. Other baselines, such as P3D, AFNO, and UNet3D, struggle significantly, exhibiting severe smoothing and checkerboard artifacts. Although the existence of implicit rotational augmentation for baselines leads to a reasonable correlation ($R^2 \geq 0.6$). ReViT's lead here demonstrates that even when data implicitly encourages symmetry learning, hard-coded symmetries achieve higher accuracy.

**3D TCF.** The Turbulent Channel Flow experiment provides the strongest evidence for the necessity of ReViT's architectural constraints with a stronger anisotropy and a limited implicit data augmentation. As shown in Table 2, ReViT performs the best with an MSE of $0.21 \times 10^{-2}$ and an $R^2$ of 0.96. This represents a reduction in error of 65% compared to the next best performing models. Notably, the global Fourier-based AFNO produces large errors ($R^2 < 0$) due to the non-periodic wall boundaries violating spectral assumptions. As illustrated in Figure 7, the turbulent structures near the no-slip walls require capturing precise anisotropic correlations. ReViT, by enforcing equivariance mathematically, successfully resolves these fine-scale turbulent streaks

*Table 2.* Performance comparison on the `MHD` and `TCF` datasets. Metrics are computed over the full chiral octahedral group $O$ with three different seeds and reported as mean $\pm$ standard deviation.

| | MHD | | TCF | |
|---|---|---|---|---|
| MODEL | MSE ($\times 10^{-2}$) ↓ | $R^2$ ↑ | MSE ($\times 10^{-2}$) ↓ | $R^2$ ↑ |
| AFNO | $16.40 \pm 42.30$ | $0.60 \pm 1.00$ | $28.40 \pm 56.40$ | $-3.79 \pm 9.49$ |
| P3D | $10.20 \pm 6.24$ | $0.73 \pm 0.15$ | $5.72 \pm 2.94$ | $0.04 \pm 0.05$ |
| UNET3D | $3.64 \pm 0.93$ | $0.90 \pm 0.03$ | $7.12 \pm 3.67$ | $-0.20 \pm 0.62$ |
| SWIN3D | $3.58 \pm 1.19$ | $0.90 \pm 0.03$ | $0.60 \pm 0.22$ | $0.90 \pm 0.04$ |
| AVIT | $2.20 \pm 0.36$ | $0.94 \pm 0.01$ | $0.60 \pm 0.23$ | $0.90 \pm 0.04$ |
| **REVIT-3D (OURS)** | $\mathbf{0.82 \pm 0.00}$ | $\mathbf{0.98 \pm 0.00}$ | $\mathbf{0.21 \pm 0.00}$ | $\mathbf{0.96 \pm 0.00}$ |

without the blurring observed in P3D or the structural noise seen in UNet3D. In 3D, ReViT achieves faster inference (12.10 ms) than the comparable transformer baselines P3D (16.45 ms) and Swin3D (13.15 ms), while delivering 96.3% and 65% lower MSE on TCF, respectively. A detailed efficiency breakdown and an analysis of ReViT's architectural variants are provided in Appendices N and P.

## 6. Discussion and Conclusions

We have introduced ReViT, a novel Vision Transformer architecture that enforces strict rotational equivariance for grid-based physical fields. By leveraging local canonicalization and invariant embeddings, ReViT overcomes the structural limitations of standard Transformers, achieving state-of-the-art performance on complex PDE benchmarks.

Despite these advancements, our approach inherits limitations of the underlying grid structure. Strictly exact equivariance is theoretically limited to the discrete symmetry group of the grid due to inevitable discretization artifacts and interpolation aliasing. While our experiments on `KF` demonstrate that data augmentation helps the model dampen the impact of these artifacts, it cannot fully eliminate them; thus, equivariance to arbitrary continuous rotations remains approximate.

Nonetheless, our results demonstrate that incorporating physical symmetries as a hard constraint significantly improves generalization and maintains physical consistency. Leveraging these capabilities in scalable Transformer architectures bears promise for a broad range of future learning tasks.

## Acknowledgments

The authors are grateful for constructive discussions with Luca Guastoni, Felix Köhler, Benjamin Holzschuh, Chengyun Wang, Yunjia Yang, Qiang Liu, and Xiyu Huang. The authors gratefully acknowledge the scientific support and HPC resources provided by the Erlangen National High Performance Computing Center (NHR@FAU) of the Friedrich-Alexander-Universität Erlangen-Nürnberg (FAU) under the NHR project b278bb. NHR funding is provided by federal and Bavarian state authorities. The authors also acknowledge the EuroHPC Joint Undertaking for providing access to the EuroHPC supercomputer LEONARDO, hosted by CINECA (Italy) and the LEONARDO consortium.

## Impact Statement

We contribute to the field of physical simulations by proposing a rotational equivariant architecture that aligns machine learning models with physical reality. This approach offers broad societal advantages, such as accelerating climate simulations to better predict environmental shifts and enhancing aerospace safety. Given the foundational nature of this architectural improvement, we foresee no immediate negative societal consequences or ethical risks.

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

# APPENDIX

This appendix provides supplementary data, mathematical derivations, and technical details to support the analysis presented in the main text. We begin in Appendix A by providing expanded quantitative results and comprehensive visualizations for the KF and 3D symmetry robustness across the chiral octahedral group for both MHD and P3D. Appendix B follows with a systematic ablation study that identifies the necessity of each ReViT component for maintaining feature correlation.

In Appendix C, we present formal definitions, algorithms, and efficiency analyses for our local basis construction methods, specifically the Average of Vectors (AOV, Appendix C.1) and the Basis of Averages (BOA, Appendix C.2), while also covering adaptations to scalar fields (Appendix C.3) and an analysis of computational overhead (Appendix C.5).

Furthermore, we address constraints and theoretical limits in Appendix D and Appendix E, which discuss periodic boundary extensions and rotation aliasing in discrete lattices, respectively. Appendix F enumerates the specific experimental configurations, hyperparameters, and hardware environments utilized for all benchmarks, ranging from rotMNIST to complex 3D TCF. This section also provides details of the baseline models compared to, such as parameter counts.

Additionally, Appendix G provides a formal error bound for approximate equivariance, Appendix H details the equivariance evaluation metric, Appendix I discusses the advantages of local over global bases, Appendix J covers extensions to tensor fields and sensitivity to reference field choice, Appendix K analyzes computational complexity, Appendix L presents a data scaling experiment, and Appendix P provides a systematic analysis of how ReViT's decoder, encoder aggregation, and window size impact efficiency and accuracy.

## A. Results details

This section provides additional quantitative data and qualitative visualizations to support the analysis of rotation equivariance and robustness of our ReViT presented in Section 5.

### A.1. Data Augmentation for 2D Kolmogorov-Flow Prediction

To ensure the statistical significance of our findings, we provide the full MSE metrics for all model variants evaluated on the KF dataset. Table 3 summarizes the mean squared error (MSE) across 20 rollout steps, averaged over three independent training runs with different random seeds.

ReViT is compared to a PDE-Transformer architecture that closely matches its structure, but omits all extensions for rotational equivariance. Thus, any differences in performance can be directly attributed to the proposed extensions. Data augmentation is indicated by parentheses, i.e., models denoted by $(+\frac{\pi}{3})$ are trained with all original data as well as the full dataset rotated by $+\frac{\pi}{3}$ as data augmentation.

The results demonstrate that our proposed equivariant architecture ReViT consistently outperforms the standard PDETrans baselines, even when the latter are bolstered by extensive data augmentation. Notably, adding a small amount of diverse angular data $(+\frac{\pi}{3}, +\frac{\pi}{6})$ allows the equivariant model to achieve its lowest error $(0.0803 \pm 0.0015)$, whereas the same data causes the baseline model to fluctuate or overfit, failing to significantly improve its baseline performance of $\approx 0.21$.

Furthermore, we provide detailed rollout visualizations for various angular pairs in Figure 8. These visualizations illustrate the model's predictive stability across the full spectrum of rotations discussed in the main text.

*Table 3.* KF2D Results Summary: Comparison of MSE ($\times 10^{-x}$, if applicable) across different model variants. Results represent the mean and standard deviation across three random initializations. Bold values indicate the best overall performance.

| Model Variant | MSE (20-step Rollout) |
|---|---|
| ReViT | $0.1361 \pm 0.0010$ |
| ReViT $(+\frac{\pi}{3})$ | $0.1031 \pm 0.0046$ |
| **ReViT** $(+\frac{\pi}{3}, +\frac{\pi}{6})$ | $\mathbf{0.0803 \pm 0.0015}$ |
| PDETrans-Aug | $0.1853 \pm 0.0138$ |
| PDETrans | $0.2125 \pm 0.0102$ |
| PDETrans $(+\frac{\pi}{3})$ | $0.2256 \pm 0.0084$ |
| PDETrans $(+\frac{\pi}{3}, +\frac{\pi}{6})$ | $0.2065 \pm 0.0058$ |

## A.2. Symmetry Robustness across the Chiral Octahedral Group

While the main text focuses on specific $2\pi$ rotations to demonstrate performance, this section provides the complete qualitative results for the chiral octahedral group ($O$). This group contains the 24 rotations that map a cube to itself, representing all possible $90°$ orientations in 3D space.

For each of the three evaluations, MHD Velocity, MHD Magnetic, and TCF Velocity, we visualize the model predictions against the ground truth for every element $g \in O$. These visualizations serve to verify that ReViT's mathematical equivariance holds strictly across all discrete rotations, maintaining physical features that baselines often distort.

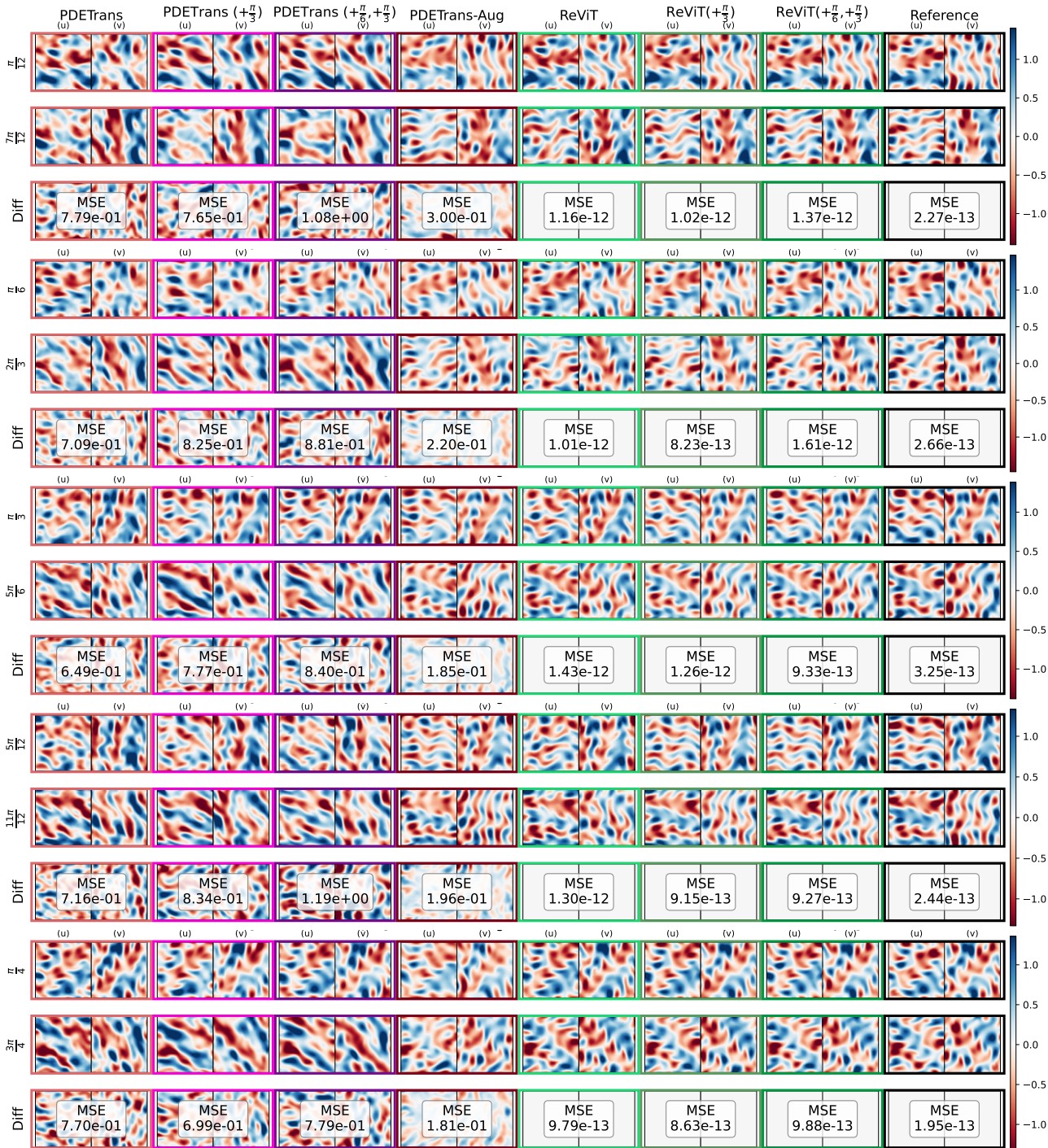

*Figure 8.* Visual comparison of model predictions across arbitrary orthogonal angular pairs $(\theta, \theta + \pi/2)$. Each row visualizes the rollout for a specific angle pair. These comparisons qualitatively confirm the numerical findings in Figure 5, specifically highlighting ReViT's ability to maintain structural coherence and rotational equivariance compared to the blurry or inconsistent outputs of the data-augmented baselines.

## B. Ablation study on Equivariance

To validate the architectural design of ReViT, we perform a systematic ablation study and a stage-by-stage analysis. This evaluation tracks the propagation of physical features through the hierarchical ReViT to precisely identify where rotational

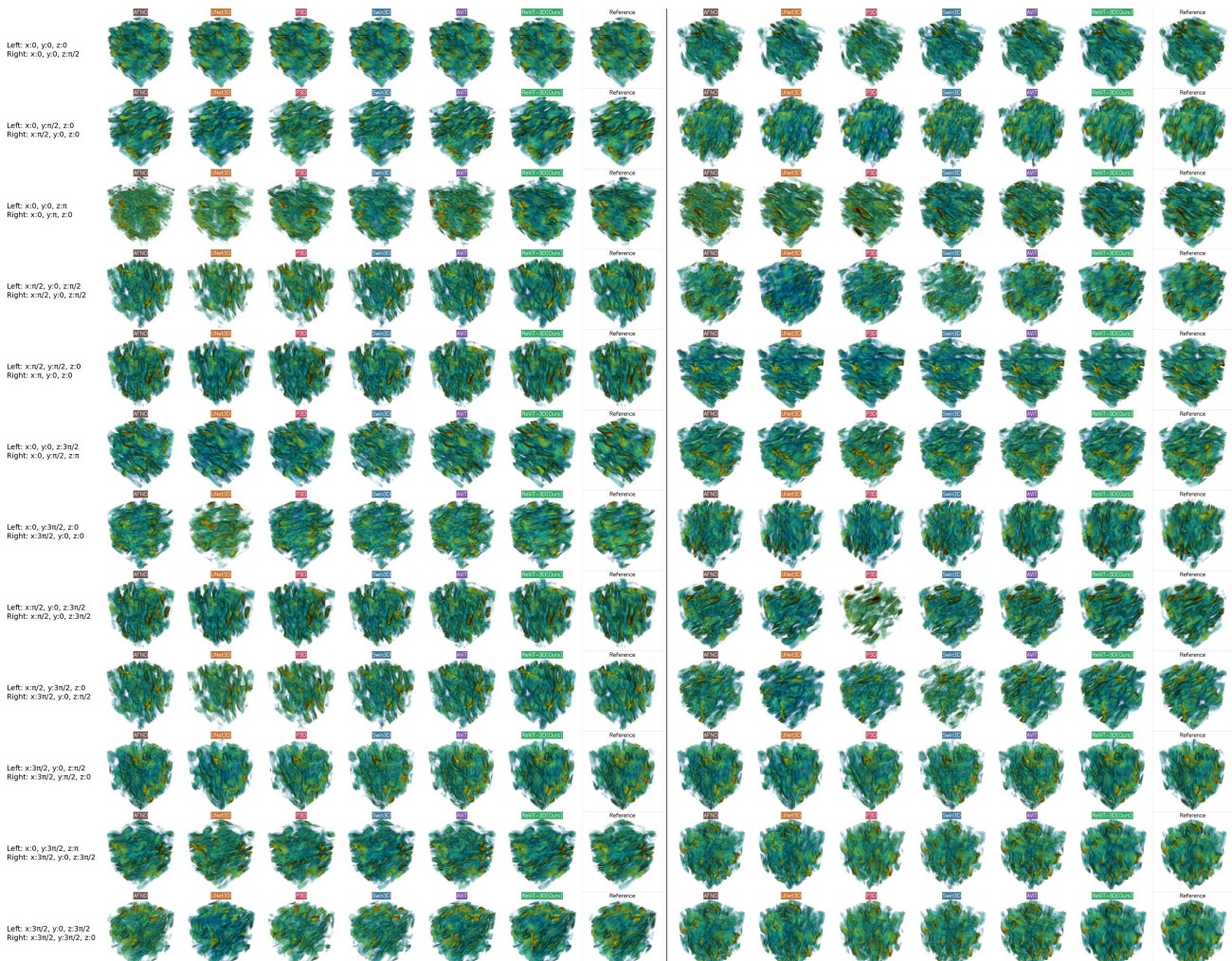

*Figure 9.* Visualization of the velocity field of `MHD` across all 24 rotations in $O$. ReViT demonstrates perfect consistency, preserving the details of turbulence across every orientation. In contrast, standard architectures like P3D and AFNO exhibit sensitivity to the orientation, with some rotations resulting in significant loss of detail and the emergence of axis-aligned artifacts.

symmetry degrades. We measure equivariance by calculating the $R^2$ correlation between the feature maps of a rotated input and the transformed feature maps of the original input. Specifically, for a network mapping $f$ and a discrete rotation $g \in O$, we evaluate the similarity between the two pathways:

$$\text{Corr}\left(f(g \cdot \mathbf{u}), g \cdot f(\mathbf{u})\right)$$

where 1.0 represents perfect equivariance. We compare the full ReViT architecture against three distinct configurations:

- *ReViT - Inv. Embed.*: Replaces the invariant embedding ( Sections 4.1 and 4.2) with standard CNN-based tokenization.

- *ReViT - Spatial Eq.*: Replaces rebased relative positional encodings (Section 4.3) with standard relative positional encodings.

- *ReViT - Eq. Decoder*: Replaces the equivariant decoder (Section 4.4) with a standard CNN-based reconstruction head.

As shown in Figure 12, the baseline ReViT maintains a strict correlation of 1.0 throughout the entire pipeline, confirming its mathematical robustness across hierarchical scales. In contrast, the removal of the invariant embedding (denoted in green) leads to an immediate collapse in correlation before the first encoding stage (Enc0). This failure, which aligns with

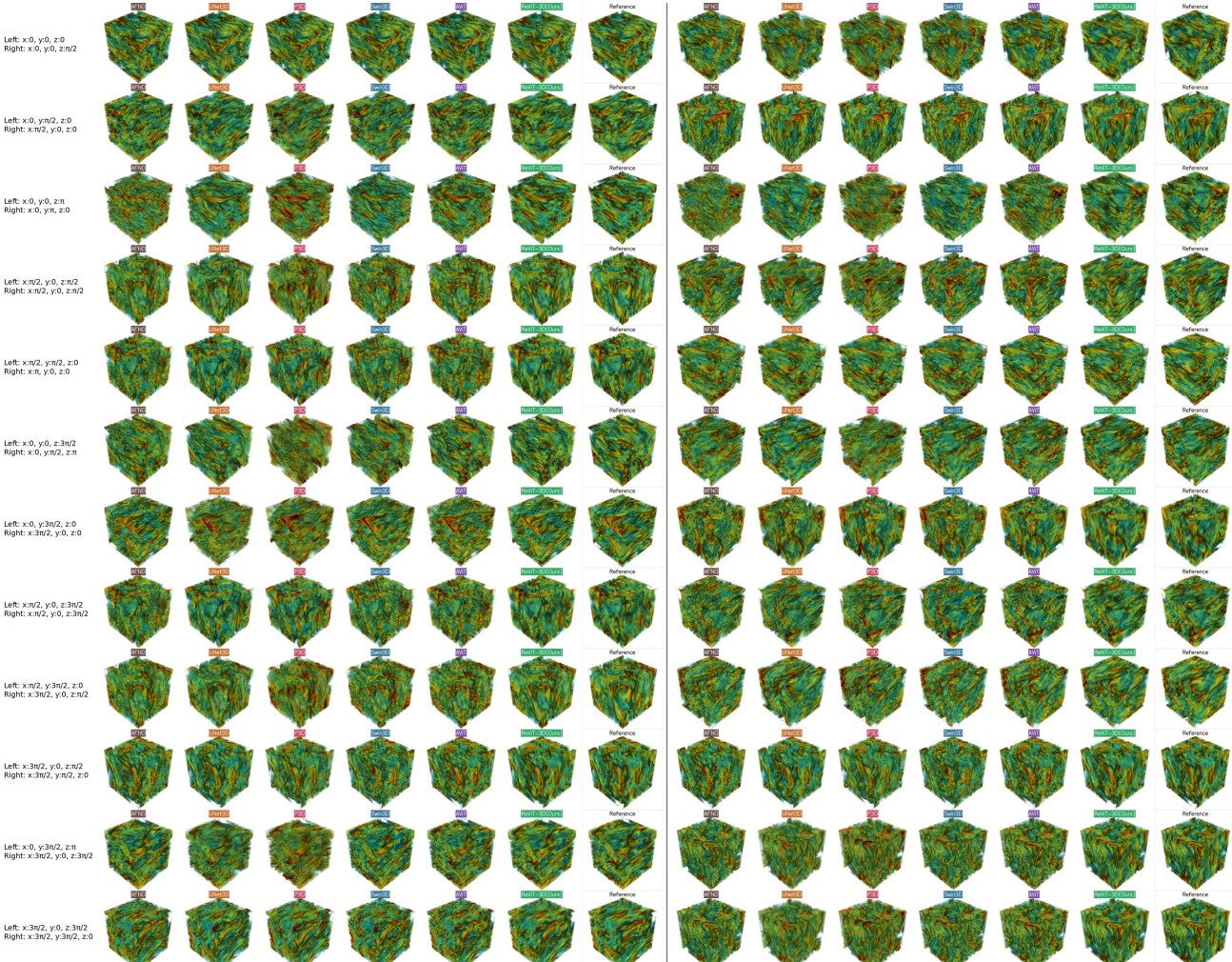

*Figure 10.* Visualization of the magnetic field structures of `MHD` across all 24 rotations in $O$. Similar to velocity results, ReViT consistently reconstructs the magnetic field with high fidelity. Baselines exhibit varying degrees of noise and structural blurring, particularly in rotations involving multi-axis flips.

Challenges 1 and 3 detailed in Section 3.1, demonstrates that standard encoders cannot handle the representational mismatch of vector fields, causing the model to lose physical consistency before feature extraction even begins.

A more nuanced degradation is observed in the *ReViT - Spatial Eq.* configuration (denoted in orange). While the model initially maintains high correlation due to the invariant embedding, the error arises in the first layer of encoding and accumulates as the signal propagates through deeper layers, confirming the necessity of addressing Challenge 2. This drift occurs because standard relative positional encodings inject a spatial bias that is inconsistent with the orientation of local canonical frames. This cumulative error eventually destroys the representational consistency within the latent space.

Finally, the ablation of the Equivariant Decoder reveals a reconstruction failure. Although the model maintains perfect internal invariance throughout the backbone, it fails to preserve full equivariance at the final output stage. This underscores that strict equivariance requires an unbroken chain: local invariant tokens must be explicitly "projected" back into the global system via equivariant reconstruction to ensure the final vector field respects the underlying physics.

## C. Local Bases

In the following, we explain and evaluate two methods that can be used in ReViT to compute $\mathbf{B}_i$, which respect the underlying physics.

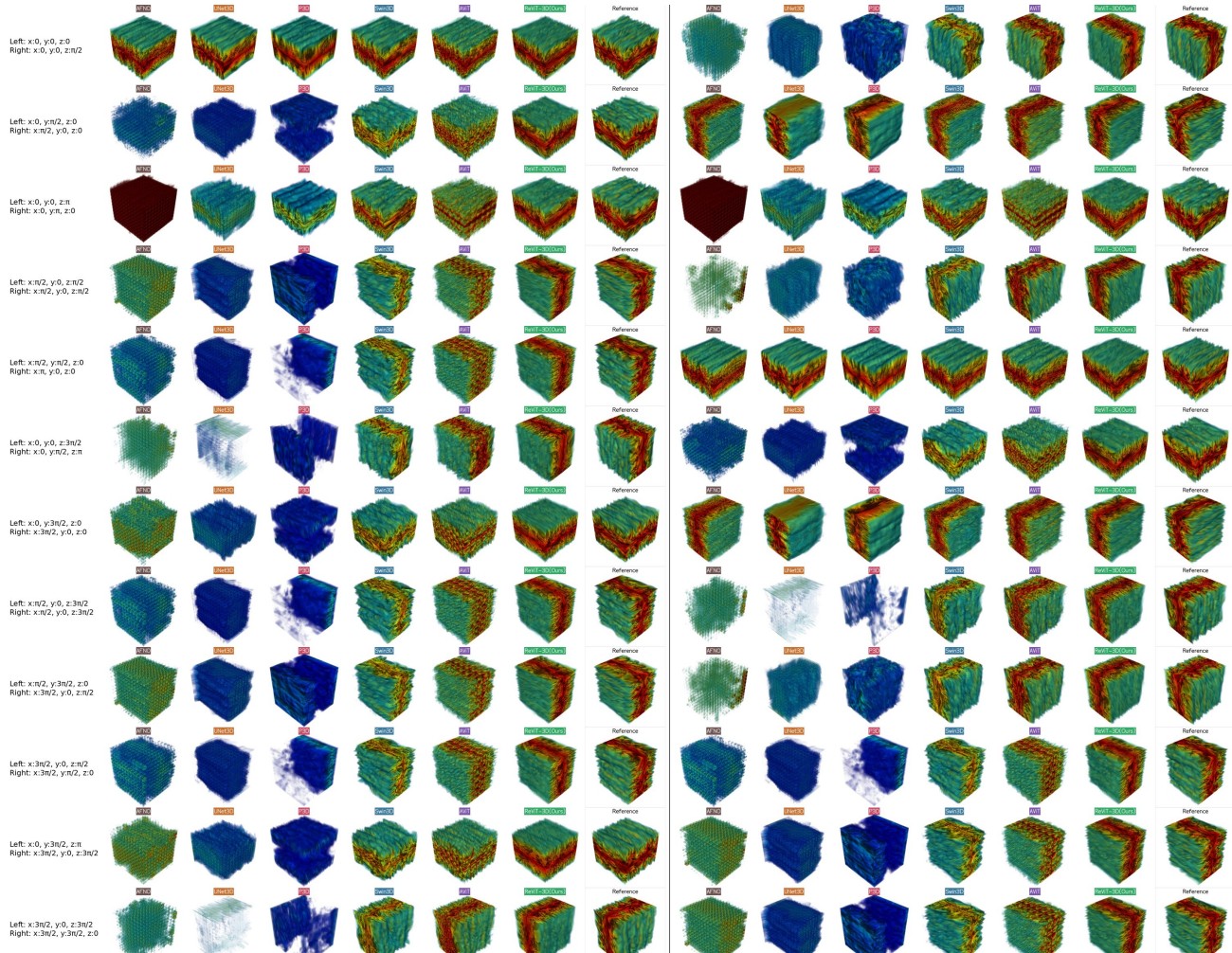

*Figure 11.* Reconstruction of velocity fields in the wall-bounded turbulence case `TCF` across all 24 rotations. This represents the most challenging test due to the high anisotropy of the flow. ReViT faithfully recovers the near-wall streaks and boundary layer gradients regardless of the channel's orientation. Other backbones like UNet3D and Swin3D show significant performance degradation in specific rotations, often failing to maintain the continuity of the streaks or incorrectly resolving the wall-normal velocity profiles.

### C.1. AOV: Average of Vectors

This method constructs a local coordinate system based on the macroscopic transport direction of the flow. This method is computationally efficient and deterministically unique, as the basis orientation is strictly defined by the flow direction and the right-hand rule, eliminating the sign and ordering ambiguities inherent to eigenvector analysis. For 3D fields, we employ a stabilized analytical orthogonalization strategy, which explicitly handles geometric singularities where the standard definition might degenerate.

#### C.1.1. 2D Basis Reconstruction

In two dimensions, the construction is straightforward. Let $\bar{\mathbf{u}}_i = [\bar{u}_x, \bar{u}_y]^T$ be the average velocity vector for patch $i$. The primary basis vector $\mathbf{b}_1$ is the normalized flow direction, and the secondary vector $\mathbf{b}_2$ is its orthonormal complement.

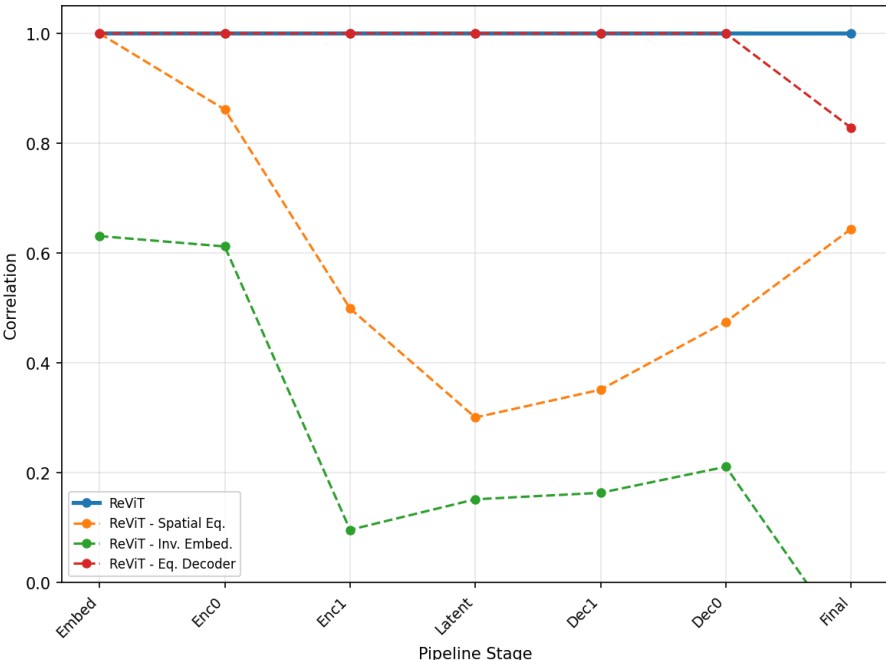

*Figure 12.* Stage-by-stage correlation analysis. The plot illustrates the degradation of equivariance ($R^2$) across hierarchical stages for different ablation configurations under a $\frac{\pi}{2}$ rotation.

Given the norm $n = \|\bar{\mathbf{u}}_i\|$, the basis is defined as:

$$\mathbf{b}_1 = \frac{1}{n} \begin{bmatrix} \bar{u}_x \\ \bar{u}_y \end{bmatrix}, \tag{8}$$

$$\mathbf{b}_2 = \frac{1}{n} \begin{bmatrix} -\bar{u}_y \\ \bar{u}_x \end{bmatrix}. \tag{9}$$

This construction guarantees a right-handed coordinate system ($\det(\mathbf{B}) = 1$). If the flow is stationary ($n < \epsilon$), we default to the global identity basis.

### C.1.2. 3D BASIS RECONSTRUCTION (AOV)

In three dimensions, defining a unique frame solely from a single vector $\bar{\mathbf{u}}_i$ is insufficient because the rotation around that vector is undefined. We resolve this by utilizing the patch vorticity $\bar{\boldsymbol{\omega}}_i$ as a secondary reference to lock the rotation angle. To avoid iterative orthogonalization schemes like QR-decomposition, we employ a direct cross-product construction.

**Primary Axis:** The first basis vector aligns with the translation direction:

$$\mathbf{b}_1 = \frac{\bar{\mathbf{u}}_i}{\|\bar{\mathbf{u}}_i\|}. \tag{10}$$

**Planar Orientation:** The third basis vector $\mathbf{b}_3$ is defined as the normal to the plane spanned by velocity and vorticity. We compute the cross product $\mathbf{c} = \bar{\mathbf{u}}_i \times \bar{\boldsymbol{\omega}}_i$.

$$\mathbf{b}_3 = \frac{\mathbf{c}}{\|\mathbf{c}\|}. \tag{11}$$

Once $\mathbf{b}_1$ and $\mathbf{b}_3$ are fixed, the second vector $\mathbf{b}_2$ is determined by the right-hand rule to complete the orthonormal triad:

$$\mathbf{b}_2 = \mathbf{b}_3 \times \mathbf{b}_1. \tag{12}$$

### C.1.3. SINGULARITY AND FALLBACK HANDLING

While AOV is generally robust, two specific singularities must be addressed:

1. **Quiescent Flow ($\|\bar{\mathbf{u}}_i\| \approx 0$):** If the mean velocity is zero, no direction is defined. In this case, $\mathbf{B}_i = \mathbf{I}$.

2. **Collinear/Beltrami Flows ($\bar{\mathbf{u}}_i \parallel \bar{\boldsymbol{\omega}}_i$):** In helical flows where velocity and vorticity are parallel (or if vorticity is zero), the cross product $\mathbf{c} = \bar{\mathbf{u}}_i \times \bar{\boldsymbol{\omega}}_i$ vanishes, leaving $\mathbf{b}_3$ undefined.

To ensure numerical stability, we implement a deterministic *Axis Fallback* procedure. If $\|\mathbf{c}\| < \epsilon$, we attempt to define the normal vector by crossing $\bar{\mathbf{u}}_i$ with the global X-axis ($\mathbf{e}_x$). If $\bar{\mathbf{u}}_i$ happens to be aligned with $\mathbf{e}_x$, we fall back to the global Y-axis ($\mathbf{e}_y$).

$$\mathbf{c}_{\text{fallback}} = \begin{cases} \bar{\mathbf{u}}_i \times \mathbf{e}_x & \text{if } \|\bar{\mathbf{u}}_i \times \mathbf{e}_x\| > \epsilon, \\ \bar{\mathbf{u}}_i \times \mathbf{e}_y & \text{otherwise.} \end{cases} \tag{13}$$

This guarantees that a valid basis is generated for any non-zero vector field.

### C.1.4. ALGORITHM

The construction procedure for AOV is summarized in Algorithm 1.

## C.2. BOA: Basis of Averages

As the strain tensor was motivated in previous work as an especially well-suited quantity to establish equivariance in fluids (List et al., 2025), we employ the following algorithm to transfer this idea to grid-based Transformer architectures. This method constructs a local coordinate system derived from the mean deformation characteristics of the fluid within a specific patch. This section details the mathematical reconstruction for 3D and 2D fields, the uniqueness constraints required to resolve eigen-ambiguities, and the computational algorithm.

### C.2.1. STRAIN-BASED BASIS RECONSTRUCTION

The core physical quantity underpinning BOA is the strain rate tensor, which describes the rate of deformation in the fluid. For a velocity field $\mathbf{u}(\mathbf{x})$, the strain rate tensor $\mathbf{S}$ at any grid point $k$ is defined as the symmetric component of the velocity gradient:

$$\mathbf{S}_k = \frac{1}{2} \left( \nabla \mathbf{u}_k + (\nabla \mathbf{u}_k)^T \right). \tag{14}$$

To obtain a basis representative of a local region (patch) rather than a single point, we partition the domain into $N$ non-overlapping patches $P_i$. We compute the *representative strain tensor* $\bar{\mathbf{S}}_i$ for patch $i$ by averaging the tensors of all constituent voxels (or pixels) $k \in P_i$:

$$\bar{\mathbf{S}}_i = \frac{1}{|P_i|} \sum_{k \in P_i} \mathbf{S}_k, \tag{15}$$

where $|P_i|$ is the volume (or area) of the patch (e.g., $P^3$ in 3D). Since $\bar{\mathbf{S}}_i$ is a real symmetric matrix, it admits an orthogonal eigendecomposition:

$$\bar{\mathbf{S}}_i = \mathbf{V}_i \boldsymbol{\Lambda}_i \mathbf{V}_i^T, \tag{16}$$

where the columns of $\mathbf{V}_i$ are the eigenvectors $\{\mathbf{v}_1, \mathbf{v}_2, \dots\}$. While these eigenvectors define principal axes of deformation, they differ from a valid coordinate basis in two ways: (1) sign ambiguity: the sign of an eigenvector is indeterminate, and (2) ordering ambiguity: the ordering of eigenvalues is not strictly coupled to a spatial orientation. To ensure the basis is unique, continuous, and rotationally equivariant, we apply a deterministic canonicalization procedure.

### C.2.2. CANONICAL BASIS UNIQUENESS PROCEDURE

We resolve ambiguities by leveraging macroscopic reference fields, the patch-averaged velocity $\bar{\mathbf{u}}_i$ and, in 3D, the patch-averaged vorticity $\bar{\boldsymbol{\omega}}_i = \frac{1}{|P_i|} \sum (\nabla \times \mathbf{u}_k)$.

---

**Algorithm 1** Average of Vector (AOV) Construction

---

**Require:** Velocity field $\mathbf{U}$, Patch size $P$, Threshold $\epsilon$.
**Ensure:** Local Basis field $\mathbf{B}$.
 1: **Step 1: Compute Means**
 2: **for** each patch $i$ **do**
 3:     $\bar{\mathbf{u}}_i \leftarrow \mathrm{mean}(\mathbf{u} \in P_i)$
 4:     $n_u \leftarrow \|\bar{\mathbf{u}}_i\|$
 5:     *// Handle Zero Velocity Singularity*
 6:     **if** $n_u < \epsilon$ **then**
 7:         $\mathbf{B}_i \leftarrow \mathbf{I}$
 8:         **continue**
 9:     **end if**
10:     **Step 2: Basis Construction**
11:     $\mathbf{b}_1 \leftarrow \bar{\mathbf{u}}_i / n_u$
12:     **if** Dimension == 2 **then**
13:         $\mathbf{b}_2 \leftarrow [-\mathbf{b}_1^{(y)}, \mathbf{b}_1^{(x)}]^T$
14:         $\mathbf{B}_i \leftarrow [\mathbf{b}_1, \mathbf{b}_2]$
15:     **else** {Dimension == 3}
16:         Compute vorticity $\bar{\boldsymbol{\omega}}_i$
17:         Calculate normal: $\mathbf{c} \leftarrow \bar{\mathbf{u}}_i \times \bar{\boldsymbol{\omega}}_i$
18:         $n_c \leftarrow \|\mathbf{c}\|$
19:         *// Handle Collinear/Beltrami Singularity*
20:         **if** $n_c < \epsilon$ **then**
21:             $\mathbf{c} \leftarrow \bar{\mathbf{u}}_i \times \mathbf{e}_x$ {Fallback to X-axis}
22:             **if** $\|\mathbf{c}\| < \epsilon$ **then**
23:                 $\mathbf{c} \leftarrow \bar{\mathbf{u}}_i \times \mathbf{e}_y$ {Fallback to Y-axis}
24:             **end if**
25:             $n_c \leftarrow \|\mathbf{c}\|$
26:         **end if**
27:         $\mathbf{b}_3 \leftarrow \mathbf{c}/n_c$
28:         $\mathbf{b}_2 \leftarrow \mathbf{b}_3 \times \mathbf{b}_1$
29:         $\mathbf{B}_i \leftarrow [\mathbf{b}_1, \mathbf{b}_2, \mathbf{b}_3]$
30:     **end if**
31: **end for**

---

**3D Reconstruction Strategy**    In three dimensions, the basis $\mathbf{B}_i = [\mathbf{b}_1, \mathbf{b}_2, \mathbf{b}_3]$ is constructed via a sequential alignment process:

1. **Primary Axis Alignment ($\mathbf{b}_1$):** We sort the raw eigenvectors $\{\mathbf{v}_1, \mathbf{v}_2, \mathbf{v}_3\}$ based on the magnitude of their projection onto the primary reference vector $\bar{\mathbf{u}}_i$. The eigenvector with the largest projection is selected as the candidate for $\mathbf{b}_1$. Its direction is fixed by enforcing a positive dot product:

$$\mathbf{b}_1 = \text{sign}(\mathbf{v}_{(1)} \cdot \bar{\mathbf{u}}_i)\, \mathbf{v}_{(1)}. \tag{17}$$

2. **Secondary Planar Alignment ($\mathbf{b}_2$):** The remaining two eigenvectors lie in the plane orthogonal to $\mathbf{b}_1$. To orient this plane, we project the secondary reference vector (vorticity) onto it:

$$\mathbf{h}_{\text{proj}} = \bar{\boldsymbol{\omega}}_i - (\bar{\boldsymbol{\omega}}_i \cdot \mathbf{b}_1)\mathbf{b}_1. \tag{18}$$

We select the eigenvector best aligned with $\mathbf{h}_{\text{proj}}$ as the candidate for $\mathbf{b}_2$ and fix its sign such that $\mathbf{b}_2 \cdot \mathbf{h}_{\text{proj}} > 0$.

3. **Chirality Enforcement ($\mathbf{b}_3$):** The final vector is determined by the right-hand rule to strictly enforce a determinant of $+1$:

$$\mathbf{b}_3 = \mathbf{b}_1 \times \mathbf{b}_2. \tag{19}$$

**2D Reconstruction Strategy**    In two dimensions, the procedure is simplified as there is no rotational plane orthogonal to the primary axis.

1. **Primary Alignment:** Similar to 3D, eigenvectors are sorted by alignment with $\bar{\mathbf{u}}_i$, and $\mathbf{b}_1$ is sign-corrected to point in the direction of flow.

2. **Chirality Enforcement:** The second basis vector $\mathbf{b}_2$ is the remaining eigenvector. We verify the chirality of the resulting matrix. If $\det([\mathbf{b}_1, \mathbf{b}_2]) < 0$, we flip the sign of $\mathbf{b}_2$ to ensure a right-handed coordinate system.

### C.2.3. SINGULARITY HANDLING

In regions where the fluid is quiescent or the deformation is isotropic (e.g., uniform flow), the eigenvectors of $\bar{\mathbf{S}}_i$ or the reference vectors may vanish, leading to numerical instability. We employ a rigorous singularity check: if the Frobenius norm of $\bar{\mathbf{S}}_i$, $\bar{\mathbf{u}}_i$, or $\bar{\boldsymbol{\omega}}_i$ falls below a threshold $\epsilon$, the basis is deemed ambiguous, and we default to the identity matrix $\mathbf{I}$. This ensures the pipeline remains robust for all flow regimes.

### C.2.4. ALGORITHM

The comprehensive procedure for constructing the BOA embeddings is detailed in Algorithm 2.

### C.3. Adaptation to Scalar Fields

To deploy our rotation-equivariant framework on scalar-based classification tasks, such as `rotMNIST` dataset, we adapt the basis construction methods from vector dynamics to image processing. While the previous methods operated on vector fields $\mathbf{u}(\mathbf{x})$, here we operate on scalar intensity fields $I(\mathbf{x})$. The fundamental bridge between these domains is the spatial gradient operator $\nabla$.

### C.3.1. GRADIENT PREPROCESSING

Input images are treated as scalar fields (for grayscale, $C = 1$) or multi-channel scalar fields (for RGB, $C = 3$). For every pixel $k$ in the input image, we compute the spatial gradient vector $\mathbf{g}_k \in \mathbb{R}^2$.

$$\mathbf{g}_k = \nabla I_k = \begin{bmatrix} \frac{\partial I}{\partial x} \\ \frac{\partial I}{\partial y} \end{bmatrix}_k. \tag{20}$$

Numerically, this is approximated using first-order Sobel operators. For RGB images, gradients are computed per channel, resulting in a set of vectors $\{\nabla R_k, \nabla G_k, \nabla B_k\}$.

---

**Algorithm 2** Basis of Average (BOA) Construction

---

**Require:** Velocity field $\mathbf{U} \in \mathbb{R}^{C \times D \times H \times W}$ (or 2D equivalent), Patch size $P$, Threshold $\epsilon$.
**Ensure:** Local Basis field $\mathbf{B} \in \mathbb{R}^{N \times C \times C}$.

1: **Step 1: Microscopic Tensor Computation**
2: **for** each voxel $k$ in $\mathbf{U}$ **do**
3:     Compute strain rate tensor $\mathbf{S}_k \leftarrow \frac{1}{2}(\nabla \mathbf{u}_k + (\nabla \mathbf{u}_k)^T)$
4:     **if** 3D **then**
5:         Compute vorticity $\boldsymbol{\omega}_k \leftarrow \nabla \times \mathbf{u}_k$
6:     **end if**
7: **end for**
8: **Step 2: Patch Aggregation**
9: Partition grid into $N$ patches of size $P^C$.
10: **for** each patch $i = 1 \dots N$ **do**
11:     $\bar{\mathbf{S}}_i \leftarrow \text{mean}(\{\mathbf{S}_k\}_{k \in P_i})$
12:     $\bar{\mathbf{u}}_i \leftarrow \text{mean}(\{\mathbf{u}_k\}_{k \in P_i})$
13:     $\bar{\boldsymbol{\omega}}_i \leftarrow \text{mean}(\{\boldsymbol{\omega}_k\}_{k \in P_i})$   *(3D only)*
14: **end for**
15: **Step 3: Basis Construction Loop**
16: **for** each patch $i = 1 \dots N$ **do**
17:     Calculate norms: $n_S = \|\bar{\mathbf{S}}_i\|_F, \quad n_u = \|\bar{\mathbf{u}}_i\|, \quad n_\omega = \|\bar{\boldsymbol{\omega}}_i\|$
18:     **if** $n_S < \epsilon$ **or** $n_u < \epsilon$ **or** ($D = 3$ **and** $n_\omega < \epsilon$) **then**
19:         $\mathbf{B}_i \leftarrow \mathbf{I}$ {Singularity fallback}
20:         **continue**
21:     **end if**
22:     Decompose: $\bar{\mathbf{S}}_i \rightarrow$ Eigenvectors $\{\mathbf{v}_1, \dots, \mathbf{v}_D\}$
23:     *// Canonicalization Procedure*
24:     Sort $\{\mathbf{v}\}$ descending by alignment $|\mathbf{v} \cdot \bar{\mathbf{u}}_i|$
25:     $\mathbf{b}_1 \leftarrow \text{sign}(\mathbf{v}_{(1)} \cdot \bar{\mathbf{u}}_i) \cdot \mathbf{v}_{(1)}$
26:     **if** Dimension $D = 3$ **then**
27:         Project secondary ref: $\mathbf{h}_{\text{proj}} \leftarrow \bar{\boldsymbol{\omega}}_i - (\bar{\boldsymbol{\omega}}_i \cdot \mathbf{b}_1)\mathbf{b}_1$
28:         Identify $k = \text{argmax}_{j \in \{2,3\}} |\mathbf{v}_{(j)} \cdot \mathbf{h}_{\text{proj}}|$
29:         $\mathbf{b}_2 \leftarrow \text{sign}(\mathbf{v}_{(k)} \cdot \mathbf{h}_{\text{proj}}) \cdot \mathbf{v}_{(k)}$
30:         $\mathbf{b}_3 \leftarrow \mathbf{b}_1 \times \mathbf{b}_2$
31:     **else** {Dimension $D = 2$}
32:         $\mathbf{b}_2 \leftarrow \mathbf{v}_{(2)}$
33:         *// Ensure right-handedness*
34:         **if** $\det([\mathbf{b}_1, \mathbf{b}_2]) < 0$ **then**
35:             $\mathbf{b}_2 \leftarrow -\mathbf{b}_2$
36:         **end if**
37:     **end if**
38:     Store $\mathbf{B}_i \leftarrow [\mathbf{b}_1, \dots, \mathbf{b}_D]$
39: **end for**

---

## C.3.2. BOA: STRUCTURE TENSOR EMBEDDING

For the Basis of Average (BOA) method, the analogue to the strain rate tensor is the *Structure Tensor*. This tensor summarizes the predominant directions of the gradient within a local neighborhood, effectively capturing texture orientation and edge geometry.

**Grayscale Construction:** For a patch $P_i$, the representative structure tensor $\mathbf{T}_i$ is computed as the average outer product of the gradients:

$$\mathbf{T}_i^{\text{Gray}} = \frac{1}{|P_i|} \sum_{k \in P_i} \mathbf{g}_k \mathbf{g}_k^T. \tag{21}$$

**RGB Construction:** For color images, we employ Di Zenzo's Color Structure Tensor (Di Zenzo, 1986). This avoids cancellation effects, where opposite gradients in different channels might sum to zero by summing the outer products of each channel independently:

$$\mathbf{T}_i^{\text{RGB}} = \frac{1}{|P_i|} \sum_{k \in P_i} \left( \nabla R_k \nabla R_k^T + \nabla G_k \nabla G_k^T + \nabla B_k \nabla B_k^T \right). \tag{22}$$

**Basis Definition:** Similar to the vector formulation, $\mathbf{T}_i$ is a symmetric positive-semidefinite matrix. We perform eigendecomposition $\mathbf{T}_i = \mathbf{V}\mathbf{\Lambda}\mathbf{V}^T$ to obtain the principal axes. To enforce uniqueness, we align the eigenvectors with the *patch-averaged gradient vector* $\bar{\mathbf{g}}_i$ (summed across channels for RGB) using the same alignment and sign-flipping procedure described in Appendix C.2.

## C.3.3. AOG: AVERAGE OF GRADIENTS (FOR IMAGES)

For this test the Average of Vector (AOV) method is applied to image gradients, thus named as Average of Gradient. This assumes that the dominant orientation of a patch is defined by its first-order gradient statistics.

For a patch $i$, we compute the mean gradient vector:

$$\bar{\mathbf{g}}_i = \begin{cases} \frac{1}{|P_i|} \sum_{k \in P_i} \nabla I_k & \text{if Grayscale,} \\ \frac{1}{|P_i|} \sum_{k \in P_i} (\nabla R_k + \nabla G_k + \nabla B_k) & \text{if RGB.} \end{cases} \tag{23}$$

The local basis $\mathbf{B}_i = [\mathbf{b}_1, \mathbf{b}_2]$ is constructed by normalizing this mean vector:

$$\mathbf{b}_1 = \frac{\bar{\mathbf{g}}_i}{\|\bar{\mathbf{g}}_i\| + \epsilon}, \tag{24}$$

$$\mathbf{b}_2 = \begin{bmatrix} 0 & -1 \\ 1 & 0 \end{bmatrix} \mathbf{b}_1. \tag{25}$$

This forms a direct, computation-light equivariant frame that aligns with the primary intensity ramp of the image patch.

### C.4. Verification of Rotational Invariance

To empirically validate the equivariance of our constructed bases, we analyze the stability of the local features under rigid-body rotations. Figure 13 illustrates this process using a slice of a 3D magnetohydrodynamic (MHD) turbulence field.

The input volume is rotated by angles $\theta \in \{0°, 90°, 180°, 270°\}$ around the Z-axis. The first column (*Global*) depicts the raw velocity field $\mathbf{u}$. As expected, the vector directions transform covariantly with the grid rotation—for example, a flow structure pointing "up" at $0°$ points "left" at $90°$.

The subsequent columns display the *rebased* features, $\mathbf{u}_{\text{local}} = \mathbf{B}_i^T \mathbf{u}_{\text{global}}$, obtained via the BOA and AOV encoders. In this local reference frame, the velocity vectors are defined relative to the patch's canonical orientation rather than global coordinates. Consequently, as the domain rotates, the pattern of local vectors rotates spatially, but the vector values relative to the underlying structure remain constant.

*Table 4.* **Computational Efficiency Benchmarks.** Inference latency (ms) for BOA and AOV under varying input resolutions (left) and patch sizes (right). Results are reported as Mean $\pm$ Std. AOV consistently outperforms BOA due to its analytical basis construction, which avoids eigendecomposition.

| | 2D (ms) | | 3D (ms) | |
|---|---|---|---|---|
| Resolution | BOA | AOV | BOA | AOV |
| 32 | $1.47 \pm 0.14$ | $\mathbf{0.65 \pm 0.07}$ | $3.72 \pm 0.86$ | $\mathbf{1.71 \pm 0.26}$ |
| 64 | $1.42 \pm 0.62$ | $\mathbf{0.70 \pm 0.29}$ | $7.78 \pm 0.91$ | $\mathbf{5.88 \pm 0.52}$ |
| 96 | $1.39 \pm 0.04$ | $\mathbf{0.71 \pm 0.01}$ | $20.59 \pm 0.10$ | $\mathbf{17.65 \pm 0.19}$ |
| 128 | $1.57 \pm 0.04$ | $\mathbf{0.90 \pm 0.02}$ | $42.95 \pm 0.60$ | $\mathbf{36.21 \pm 0.66}$ |
| 160 | $1.86 \pm 0.12$ | $\mathbf{1.14 \pm 0.04}$ | $83.64 \pm 0.49$ | $\mathbf{70.99 \pm 0.54}$ |
| 256 | $2.84 \pm 0.05$ | $\mathbf{2.13 \pm 0.01}$ | — | — |

| | 2D (ms) | | 3D (ms) | |
|---|---|---|---|---|
| Patch Size | BOA | AOV | BOA | AOV |
| 1 | $2.00 \pm 0.06$ | $\mathbf{1.12 \pm 0.02}$ | — | — |
| 2 | $1.68 \pm 0.06$ | $\mathbf{0.96 \pm 0.02}$ | $54.66 \pm 0.79$ | $\mathbf{39.97 \pm 0.16}$ |
| 4 | $1.57 \pm 0.04$ | $\mathbf{0.89 \pm 0.01}$ | $43.61 \pm 0.56$ | $\mathbf{36.53 \pm 0.62}$ |
| 8 | $1.57 \pm 0.08$ | $\mathbf{0.88 \pm 0.01}$ | $42.19 \pm 0.59$ | $\mathbf{36.30 \pm 0.67}$ |

We quantify this stability by calculating the Mean Squared Error (MSE) between the rebased features of the rotated field and the spatially rotated features of the original field:

$$\text{MSE} = \frac{1}{N} \sum \|\text{Encoder}(\mathcal{R}(\mathbf{u})) - \mathcal{R}(\text{Encoder}(\mathbf{u}))\|^2. \tag{26}$$

For both BOA and AOV, the MSE is strictly zero (within numerical precision to $10^{-7}$). This confirms that our basis construction successfully projects the vector into invariant states, allowing us to leverage the invariant network to further process.

### C.5. Analysis of Computational Efficiency

We analyze the computational overhead introduced by ReVit and the proposed basis construction modules by comparing the strain-based BOA method with the vector-averaged AOV method. All benchmarks were conducted on a single NVIDIA RTX A5000 GPU with a batch size of 1, while varying the input resolution and patch size $P$. The resulting latency measurements (i.e., ntruntime) are reported in Table 4 and visualized in Figure 14. Across all evaluated settings, AOV consistently exhibits lower inference latency than BOA, indicating superior computational efficiency.

The latency results in Table 4 and Figure 14 reveal a consistent computational advantage of AOV over BOA across all evaluated configurations. When varying the input resolution at a fixed patch size, both methods exhibit approximately linear scaling with respect to the number of pixels or voxels, reflecting the dominant cost of per-token basis construction. However, AOV consistently achieves lower inference latency at all resolutions. This gap is particularly pronounced at lower resolutions, where the fixed overhead of eigendecomposition in BOA becomes more significant as the number of patches increases. Although the relative difference narrows as resolution grows and the overall computation becomes dominated by memory access and aggregation, AOV maintains a clear performance lead even in high-throughput regimes.

This behavior can be directly attributed to the algorithmic formulation of the two methods. BOA relies on strain-rate estimation, which requires constructing a second-moment tensor by aggregating outer products over each patch, followed by an eigendecomposition of a symmetric matrix to extract principal directions. Despite the small dimensionality of the tensor, the diagonalization step introduces nontrivial arithmetic intensity, control-flow divergence, and limited opportunities for instruction-level parallelism on GPUs. In contrast, AOV is formulated entirely in terms of first-moment statistics and avoids both tensor assembly and spectral decomposition. Its basis construction relies only on elementary vector operations such as normalization and cross products, which are highly amenable to hardware-level optimization and efficient parallel execution.

A similar trend is observed when varying the patch size at a fixed resolution. Increasing the patch size reduces the total number of tokens quadratically or cubically, depending on the spatial dimensionality, leading to lower latency for both BOA and AOV. Nevertheless, AOV (as used throughout the main paper) remains uniformly faster across all tested patch sizes. While larger patches reduce the absolute latency gap by decreasing the number of basis constructions, they do not eliminate it, indicating that the efficiency gains of AOV are intrinsic to its analytical design rather than a byproduct of token count reduction or specific hyperparameter choices.

To conclude, ReViT with an AOV basis incurs a mild computational overhead that scales linearly with the number of processed input tokens. We believe that, due to the improved capabilities for generalization as reported in the main text above, this computational overhead is a price worth paying for a large number of real-world scenarios.

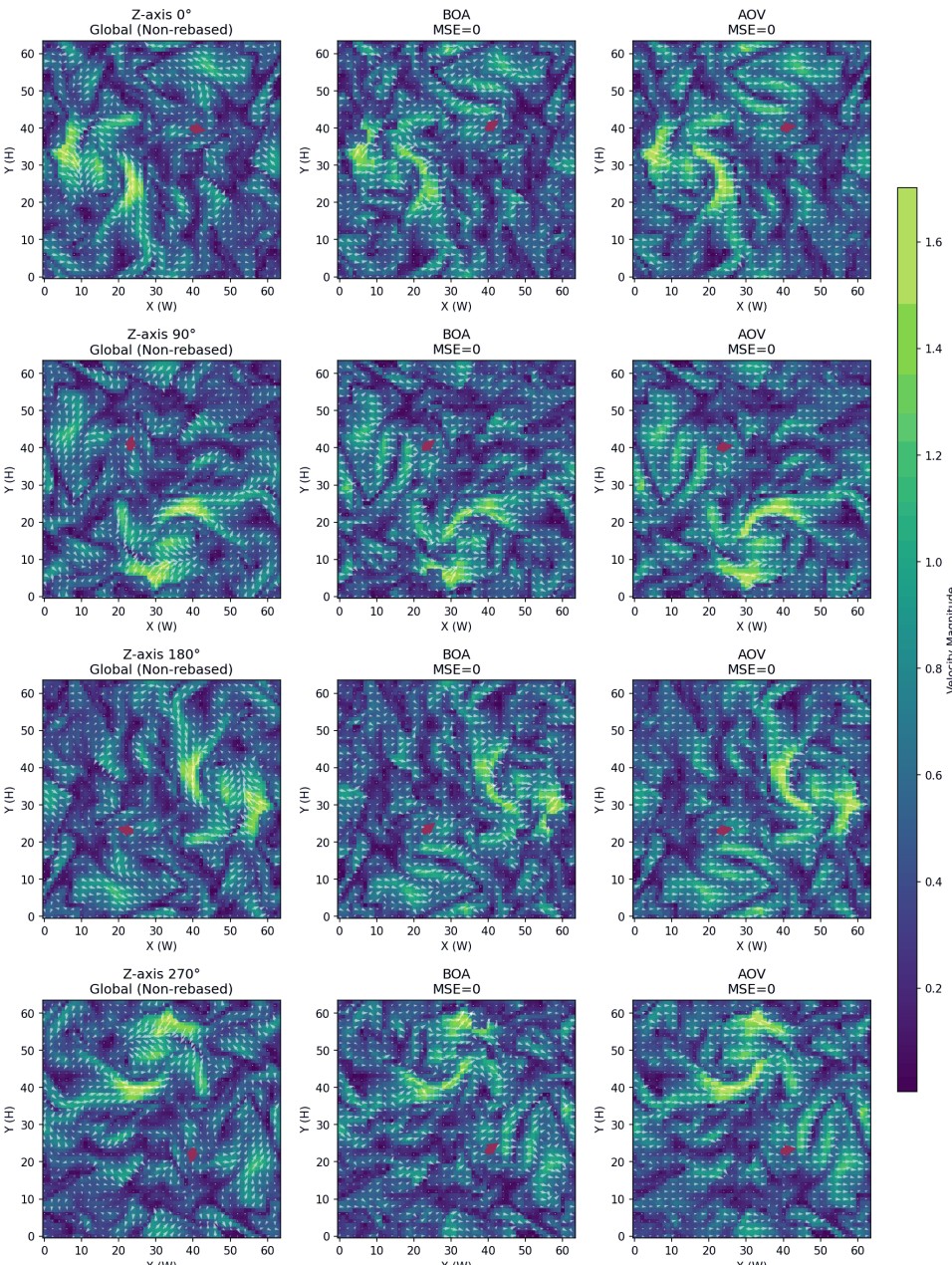

*Figure 13.* **Visualization of Rotation Invariance in 3D Rebased Features.** The rows display a snapshot of 3D MHD turbulence subjected to rotations of $\{0°, 90°, 180°, 270°\}$ around the Z-axis. The *Left* column shows the original global velocity field, where vectors rotate physically with the domain. The *Middle* (BOA) and *Right* (AOV) columns visualize the *local* feature vectors projected into the computed canonical bases. The arrow indicates the direction of the vectors, and the red arrow serves as a fiducial marker to track a specific grid for better understanding. Obviously, the direction of both BOA and AOV never changed with the rotation. The annotated MSE = 0 confirms that the local feature representations are numerically invariant to global rotation, effectively behaving as scalar fields that translate spatially without vectorial transformation.

## D. Valid Rotation Range and Periodic Boundary Extension

The rotation of a discrete two-dimensional sampled image, $f(x, y)$, defined on a rectangular grid $N \times N$, presents a fundamental geometric challenge regarding the validity of the sampling domain. As noted by Larkin et al. (Larkin et al., 1997), a square sampling grid rotated by an angle $\theta$ does not spatially overlap with the original grid, except when the rotation is a multiple of $90°$. This mismatch results in "inevitable non-overlapping corner regions" where the resampling algorithm

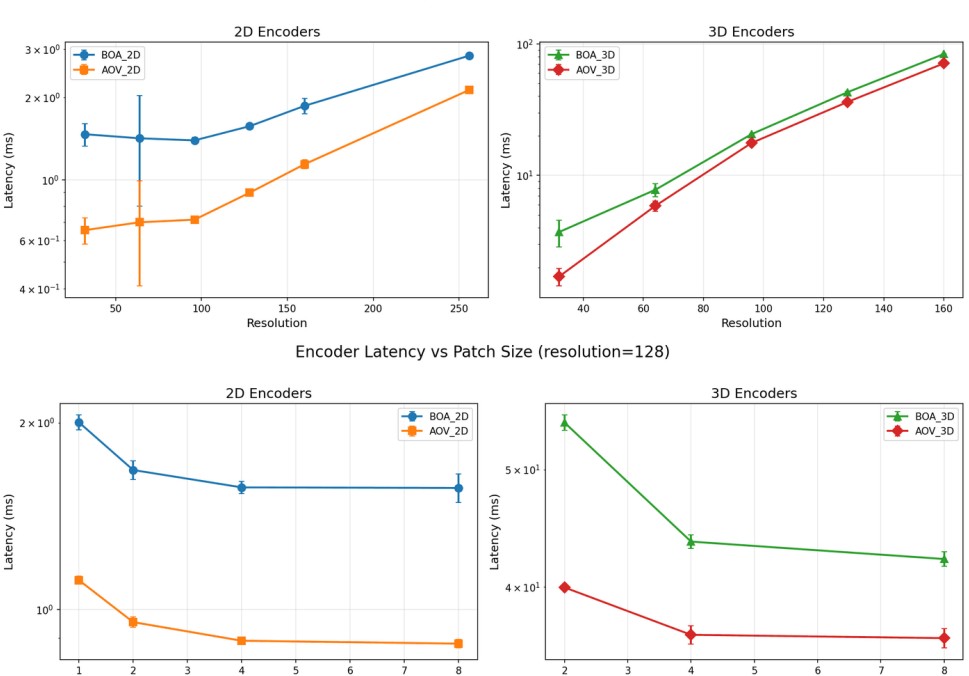

*Figure 14.* **Encoder Latency Analysis.** Comparison of BOA and AOV encoders across 2D and 3D tasks. **Left:** Latency scaling with input resolution (fixed patch size $P = 4$). AOV consistently outperforms BOA because the analytical basis construction is simpler than eigendecomposition. **Right:** Latency reduction with patch size increasing (fixed resolution 128), AOV still outperforms BOA.

attempts to query values outside the defined domain of $f(x, y)$.

### D.1. Geometric Constraints of the Sampling Grid

Strictly speaking, the valid region for a rotation by an arbitrary angle is limited to the inscribed circular rotation zone centered within the grid. Information residing in the corners of the square domain (outside this circular zone) is susceptible to truncation or aliasing artifacts during rotation.

This loss of information is illustrated in the top row of Figure 15. Using a standard rotation approach, the corner regions of the rotated frame fall into the undefined void. Consequently, when the image is rotated back to its original orientation ($-\theta$), the corner information is irretrievably lost, resulting in the blurred and distorted artifacts highlighted by the red bounding boxes.

### D.2. Periodic Domain Expansion

To address this limitation for data exhibiting periodic boundary conditions, we implement a periodic domain expansion strategy. Our approach extends the sampling domain by wrapping the field onto itself.

Given an input field of dimensions $H \times W$, the minimal canvas size required to capture the full domain at the maximal extent of rotation (occurring at $\theta = 45°$) is derived from the diagonal length of the original grid. The field is padded to a new dimension $H' \times W'$ where:

$$H' \geq H\sqrt{2}, \quad W' \geq W\sqrt{2} \tag{27}$$

This expansion ensures that the "smallest square enclosing the circular zone" (Larkin et al., 1997) is fully populated with valid data derived from the periodic neighbors.

As demonstrated in the bottom row of Figure 15, applying periodic padding prior to rotation allows the sampling grid to seamlessly capture data across the periodic boundaries. The subsequent inverse rotation and cropping operation yields a near-perfect reconstruction of the original field, effectively resolving the valid range artifacts observed in the standard

approach.

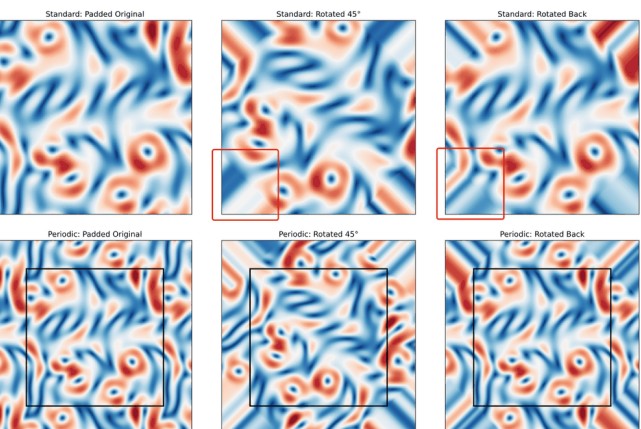

*Figure 15.* Comparison of rotation methods on a periodic scalar field. **Top Row (Standard):** Rotation without adequate domain expansion results in information loss at the corners (highlighted in red), as the sampling grid extends beyond the valid range. **Bottom Row (Periodic):** The proposed method applies periodic padding to scale the domain by $\approx \sqrt{2}$, simulating an infinite field. The black inset box indicates the original domain boundaries, showing how periodic neighbors fill the extended range, allowing for lossless recovery of the original field.

## E. Rotation Aliasing in Discrete Lattices

A significant theoretical hurdle in applying $SO(d)$ equivariance to grid-based data is the fundamental geometric incompatibility between the continuous rotation group and the discrete lattice $\mathbb{Z}^d$. As noted by Park et al. (Park et al., 2009), the rotation of a finite 2D lattice is generally not compatible with the lattice structure itself, as the grid points are not mapped to themselves under an arbitrary rotation. Consequently, a perfect rotation by an arbitrary angle $\theta$ maps integer grid coordinates to non-integer positions, necessitating the use of interpolation methods such as bilinear or bicubic resampling.

### E.1. Interpolation-Induced Information Loss

Standard rotation implementations rely on spatial interpolation (e.g., 'grid_sample' with bilinear or bicubic modes) to estimate intensity values at off-grid coordinates. However, local interpolation is an inexact process that inherently degrades global information. This degradation manifests as aliasing artifacts, where high-frequency errors are visible in the difference maps of rotated fields. Generally, it's caused by there is no mathematically exact method to rotate and resample the underlying band-limited series from which discrete samples are drawn. The resulting "loss of information" prevents the system from achieving true equivariance, as the composition of rotations (e.g., rotating forward by $\theta$ and backward by $-\theta$) fails to reconstruct the original signal exactly.

### E.2. Angle-Dependent Error Periodicity

The magnitude of rotation aliasing is highly dependent on the rotation angle, creating a characteristic error periodicity. Our empirical analysis of Mean Squared Error (MSE) across rotation angles reveals distinct minima and maxima:

- **Trivial Rotations ($k \cdot 90°$):** The error drops to zero at rotation angles that are multiples of $90°$ ($\theta \in \{0, \pi/2, \pi, \dots\}$). At these specific angles, the transformation aligns perfectly with the Cartesian grid. As Park et al. explain, rotations by multiples of $\pi/2$ can be perfectly obtained by simply reassigning pixel values to the rectangular grid, avoiding interpolation entirely. These angles provide a "perfect baseline" where exact mathematical equivariance is preserved.

- **Non-Trivial Rotations:** For arbitrary angles (e.g., $\theta = 45°$ or $30°$), the grid mismatch is maximized. The discretization error introduces peaks in the MSE, as the interpolation algorithms must approximate values between grid points. This confirms that while discrete Fourier transforms and lattice representations function well for translation, they lack the rotational invariance of their continuous counterparts.

In summary, while grid-based representations can achieve exact equivariance under the discrete subgroup $C_4$ (90-degree rotations), they suffer from unavoidable aliasing errors under the continuous group $SO(2)$, rendering exact equivariance

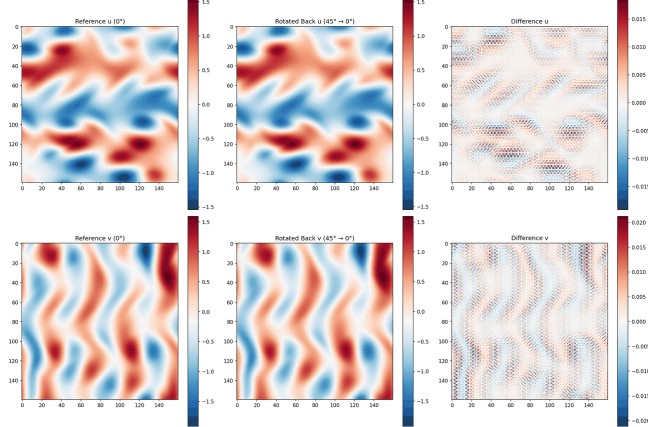

*Figure 16.* **Visualization of Rotation Aliasing in Vector Fields.** Analysis of a vector field rotated by $30°$ and then rotated back by $-30°$ using bilinear interpolation. While the reconstructed fields (*Rotated Back u, v*) visually resemble the reference, the *Difference* maps reveal significant high-frequency aliasing artifacts distributed across the spatial domain. These errors confirm the loss of information caused by the geometric incompatibility between the rotation operator and the discrete grid.

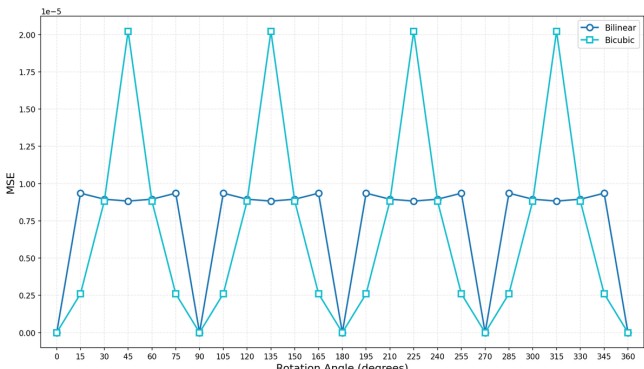

*Figure 17.* **Angle-Dependent Discretization Error.** Mean Squared Error (MSE) between the original and reconstructed fields after a forward and backward rotation, plotted across angles from $0°$ to $360°$ with step=$15°$. The error drops to zero at trivial rotations ($k \cdot 90°$), where grid alignment allows for perfect pixel reassignment. For non-trivial angles, the discretization error peaks at $45°$ offsets, with Bicubic interpolation exhibiting higher peak errors than Bilinear due to resampling overshoot.

theoretically impossible on standard discrete lattices without moving to alternative basis representations such as Hermite expansions.

## F. Experimental Details

### F.1. rotMNIST

For the rotMNIST case, we implemented the ReViT with a depth of 6 blocks and an embedding dimension of 192. For the patch embedding, we employed an AOG-based encoder with a patch size of 4. Equivariance is also maintained via Equivariant Relative Positional Encodings. The model, with 2.75 million trainable parameters, was trained for 300 epochs using the AdamW optimizer with a learning rate of 0.001 and a weight decay of 0.1. We utilized a ReduceLROnPlateau scheduler and applied a dropout rate of 0.1, with a specific attention dropout of 0.3 to prevent overfitting on the `rotMNIST` dataset.

The accuracy of the other benchmark models listed in Table 1 is reported using results from the corresponding original papers, including their trainable parameter count of 44.67 k. The time consumption and memory usage are measured by running the publicly released implementations on our hardware. Unfortunately, the baseline codebase is implemented using PyTorch 1.6.0, which requires a legacy PyTorch version. Consequently, all baseline experiments are conducted on a single NVIDIA GeForce RTX 2080 Ti GPU.

For models with large memory requirements (e.g., GAS-Nets ($R_{12}$) with batch size 32), direct training under the original batch size is not feasible on a single RTX 2080 Ti GPU. To estimate the peak memory usage under the original setting, we run the model with batch sizes of 1, 2, and 3, and record the corresponding peak GPU memory consumption during training. Since the model architecture and parameter size remain fixed, the activation memory scales approximately linearly with batch size. We therefore fit a linear regression between batch size and observed peak memory usage, and extrapolate the memory requirement for batch size 32 based on this regression. The reported memory usage corresponds to the estimated peak GPU memory consumption during training. We believe this linear extrapolation strategy is commonly adopted when direct measurement under the target batch size is infeasible due to hardware limitations, and is valid when model parameters and computation graphs remain unchanged.

A notable architectural distinction for the `rotMNIST` configuration is the use of *basis-aware self-attention*. Unlike the PDE tasks, where the input fields are spatially dense and yield well-defined canonical bases for virtually all patches, MNIST images are sparse: a large fraction of the $7 \times 7{=}49$ patches (at $P{=}4$) lie on the black background, where the spatial gradient $\nabla I$ vanishes and the basis construction degenerates. In this regime, the equivariant relative positional encoding alone cannot provide a reliable orientation signal, as the rebased displacements $\mathbf{B}_i^T(x_j - x_i)$ are less informative for degenerate bases. To compensate, we encode the basis directly into the attention mechanism: features are interpreted as groups of 2D vectors, and keys and values are rebased into the query's local frame via $\mathbf{R}_{ij} = \mathbf{B}_i^T \mathbf{B}_j$ before computing attention scores, analogous to how Rotary Position Embeddings (RoPE) (Su et al., 2024) encode relative positions through rotation matrices. This is not required for PDE tasks, where the dense, continuous physical fields ensure that the equivariant PE and hierarchical backbone already provide sufficient rotational information.

### F.2. 2D Advection

The advection dataset is derived from REMUS (Lino et al., 2022a). It features an advecting velocity field $\mathbf{u}$ that is constant across the periodic computational domain. This velocity field moves a passive scalar marker $\psi$, as described by the advection equation

$$\frac{\partial \psi}{\partial t} + \mathbf{u} \cdot \nabla \psi = 0. \tag{28}$$

These scalar and vector quantities are discretized by a randomized graph in the original dataset. Following List et al. (2025), we only consider obstacle-free advection scenarios and split the dataset into 1300 training trajectories and 200 test trajectories, each containing a regular grid using bilinear interpolation. Four comparative graphnet architectures are included in our evaluations. They represent different methods of achieving rotational equivariance in message-passing. SEGNN (Brandstetter et al., 2021) achieves rotational equivariance by encoding steerable features that are constructed from the irreducible representations of the input. REMUS (Lino et al., 2022b) projects the features along edges connecting the graph nodes and operates on these edge features as well as angle features to reconstruct vector quantities. ReGNN (List et al., 2025) is based on an approach similar to the one shown on transformers in this paper. It uses local basis transformations to construct invariant features that are transformed back to the original coordinate system at the output.

The baseline results (SEGNN, ReGNN, REMUS) are reported as in List et al. (2025), and the models were thus trained and tested on the original irregular mesh representations. The networks use 10 message-passing steps and have around 400k trainable parameters.

We also benchmark against the Lorentz-equivariant Geometric Algebra Transformer (LGATr (Spinner et al., 2025)). While L-GATr achieves exact equivariance through internal multivector representations, prior research has noted that these specialized layers can incur substantial computational costs. In our initial experiments, we evaluated a larger configuration of the L-GATr model comprising approximately 3.4 million trainable parameters. However, empirical testing on an NVIDIA A100 SXM4 64GB GPU revealed significant scalability constraints, as the high memory requirements of the multivector representations restricted the maximum batch size to 12. At this scale, training latency was found to be prohibitive, with a single epoch requiring approximately three hours to complete. Such a configuration would necessitate over 30 days of continuous compute time to reach the 300-epoch target, rendering large-scale experimentation impractical within typical resource allocations. Consequently, we pivoted to a highly optimized variant to facilitate more efficient iteration while maintaining the architectural benefits of Lorentz equivariance. This model utilizes a significantly reduced footprint of 0.034M trainable parameters. The architecture consists of two transformer blocks, each employing two attention heads. To manage the trade-off between invariant and equivariant information processing, the hidden latent space is partitioned into 16 scalar channels and 8 multivector channels. The geometric message passing within this architecture is complemented by a Geometric MLP with an expansion factor of two and scalar-gated nonlinearities, a design choice that substantially reduces

the computational overhead. Other training parameters are identical to those for training ReViT and PDETrans, detailed below.

For ReViT, we employ 3 hierarchical levels, following a depth configuration of [2, 4, 8, 4, 2] and progressively scaling the embedding dimension from 32 to 128. We utilize an AOV patch encoder with a unit patch size and a window size of 8. To ensure precise spatial reconstruction, the model features an equivariant decoder integrated with Fourier Features ($\sigma = 10.0$) and an invariant aggregation method. Training was conducted for 300 epochs with a batch size of 64 using AdamW with a learning rate of 0.001, and the model's performance was evaluated over a 20-step temporal rollout. The trainable parameter count is approximately 3.2M. The configuration for the non-equivariant benchmark, PDETrans (Holzschuh et al., 2025b), was kept similar to ReViT, but with a slightly higher parameter count of about 3.6M. All PDE-related experiments, `Adv`, `KF`, `MHD`, and `TCF`, were conducted on an NVIDIA A100 SXM4 (64GB) GPU, each evaluated using three independent random seeds.

### F.3. 2D KF

The 2D Kolmogorov Flow (`KF`) scenario within the APEBench framework models the dynamics of an incompressible fluid governed by the Navier-Stokes equations under stationary sinusoidal forcing. In this benchmark, the system is expressed using a streamfunction-vorticity formulation where the scalar vorticity field $u$ evolves over time.

$$\frac{\partial u}{\partial t} = -b\left(\begin{bmatrix} 1 \\ -1 \end{bmatrix} \odot \nabla(\Delta^{-1}u)\right) \cdot \nabla u + \nu\nabla \cdot \nabla u + \lambda u - k\cos\left(k\frac{2\pi}{L}y\right) \tag{29}$$

We calculate the velocity field $\mathbf{u} = (u_{vel}, v_{vel})$ based on the vorticity $u$ via the Poisson equation for the stream function $\psi$, defined as $\Delta\psi = u$. In Fourier space, this allows for the efficient computation of velocity components as $\hat{u}_{vel} = ik_y\hat{\psi}$ and $\hat{v}_{vel} = -ik_x\hat{\psi}$. The physical configuration utilizes a domain extent of $L = 2\pi$ and a Reynolds number $Re = 100$, where the viscosity coefficient is defined as $\nu = 1/Re$. Additional constitutive parameters include a linear drag coefficient $\lambda = -0.1$ and a forcing wavenumber $k = 4$.

To examine the performance of equivariant models on this dataset, we augment the velocity field by manually rotating the data across angular intervals of $\frac{\pi}{12}$ within the range $(0, \pi)$. While arbitrary rotation of a square sampling grid typically results in non-overlapping corner regions and information loss Appendix D, the periodic nature of the `KF` domain allows us to mitigate these artifacts. Specifically, we implement a periodic domain expansion by padding the field to a dimension of at least $H\sqrt{2} \times W\sqrt{2}$, ensuring the rotating sampling grid always queries valid data from periodic neighbors.

For both ReViT and PDETrans, we adopt the configuration used in the `Adv` setup, but with a slightly higher weight decay of 0.001 and a batch size of 12. The resulting model sizes remain comparable, with 3.2M and 3.6M trainable parameters for ReViT and PDETrans, respectively. We evaluate the impact of data augmentation by training several model variants on different subsets of rotated data. The baseline ReViT and PDETrans models are trained solely on the $\frac{\pi}{4}$ rotation. The models denoted as $(+\frac{\pi}{3})$ are trained on a combined dataset of $\frac{\pi}{4}$ and $\frac{\pi}{3}$ rotations, while the $(+\frac{\pi}{3}, +\frac{\pi}{6})$ variants utilize rotations of $\frac{\pi}{4}, \frac{\pi}{3}$, and $\frac{\pi}{6}$. Finally, we include PDETrans-Aug, which utilizes a random augmentation angle, as a common practice for data-augmented equivariant models.

### F.4. 3D MHD

The MHD dataset within the Well collection represents a large-scale simulation of magnetohydrodynamic turbulence specifically designed to model essential components of astrophysical phenomena such as galaxy formation, the solar wind, and interstellar medium (ISM) dynamics. The data provided in this collection consists of isothermal simulations without self-gravity, initially computed at a resolution of $256^3$ and subsequently downsampled to $64^3$ following anti-aliasing via an ideal low-pass filter to ensure the tasks remain approachable for modern machine learning architectures. The primary control parameters for these simulations are the dimensionless sonic Mach number ($M_s \equiv |v|/c_s$) and the Alfvénic Mach number ($M_A \equiv |v|/\langle v_A \rangle$), where $c_s$ represents the isothermal sound speed and $v_A$ denotes the Alfvén speed. In our study, we employ the $M_A = 0.7$ and $M_s = 0.5$.

The dynamics of MHD are controlled by the following system of equations for mass, momentum, and magnetic induction:

$$\frac{\partial \rho}{\partial t} + \nabla \cdot (\rho \mathbf{v}) = 0 \tag{30}$$

$$\frac{\partial (\rho \mathbf{v})}{\partial t} + \nabla \cdot (\rho \mathbf{v} \mathbf{v} - \mathbf{B} \mathbf{B}) + \nabla p_{tot} = 0 \tag{31}$$

$$\frac{\partial \mathbf{B}}{\partial t} - \nabla \times (\mathbf{v} \times \mathbf{B}) = 0 \tag{32}$$

where $\rho$ denotes the gas density, $\mathbf{v}$ represents the velocity vector field, $\mathbf{B}$ is the magnetic field vector, $p$ is the gas pressure, and $\mathbf{I}$ is the identity matrix. The simulations employ an isothermal equation of state, $p = c_s^2 \rho$, and utilize periodic boundary conditions.

The performance of ReViT is evaluated against several established architectures in the field of neural operators and physical surrogate modeling. P3D (Holzschuh et al., 2025a) is a scalable foundation model specifically engineered for high-resolution 3D physical simulations. It utilizes a memory-efficient attention mechanism combined with a hierarchical structure to capture both local and global dependencies in volumetric data, enabling it to process large-scale 3D grids that are typically prohibitive for standard Transformers. In our study, we kept a similar setup to the original paper with 11.2 million parameters, only modifying the input/output channel to fit the current case.

Similarly, the Swin3D model, our own implementation of SwinV2 (Liu et al., 2022), uses a hierarchical shifted-window approach with 14.4 million parameters and a window size of 2, enabling the capture of multi-scale turbulent features. The AViT model (McCabe et al., 2024) leverages an axial attention mechanism to maintain computational efficiency while facilitating transfer learning to downstream tasks. In our study, it consists of 11.5 million trainable parameters. The AFNO (Guibas et al., 2021) configuration comprises 13.0 million parameters. This model is an extension of the standard FNO that incorporates a token-mixing strategy within the Fourier domain. It utilizes a block-diagonal weight sharing scheme and adaptive weighting of frequency modes. We applied 8 diagonal blocks and a sparsity threshold of 0.01 to filter frequency modes in the Fourier domain. We also employ a 3D variant of UNet (Ronneberger et al., 2015), denoted by UNet3D. When the parameter count of UNet3D is too high, this architecture has difficulties to converge; thus, we decrease the parameter to the same level as ReViT-3D, to about 3.5 million. For ReViT-3D, we design the setup to ensure the parameter size can be identical with the previous ReViT, thus we decrease the depth configuration of [2, 4, 8, 4, 2] by half, resulting in a compact parameter count of approximately 3.6 million.

Every model in this suite is trained with an identical setup of `KF`, using the AdamW optimizer with a weight decay of $0.001$, a starting learning rate of $0.001$, and a ReduceLROnPlateau scheduler for a duration of 300 epochs. Unlike previous 2D cases, for 3D cases, to ensure physical consistency and focus the learning process on temporal dynamics, all models use an Euler connection, where the network predicts the time derivative of the state rather than the absolute next state. Also, for 3D, given the heterogeneity in benchmark architectures and parameter scales, we adopt mini-batch gradient descent with a batch size of 1 for all models to maintain uniform memory feasibility.

### F.5. 3D TCF

The 3D Turbulent Channel Flow (`TCF`) dataset serves as a canonical benchmark for wall-bounded turbulence. This configuration represents a periodic channel with physical walls along the $y$-axis where no-slip boundary conditions are enforced. The flow is driven by a dynamic forcing term to maintain energy levels against the dissipation caused by wall friction. In this dataset, we consider a Reynolds number of $Re = 400$, with 200 snapshots captured at a temporal resolution of $\Delta t = 0.1$. The simulation is discretized on a $192 \times 96 \times 96$ spatially adaptive grid, which features a refined resolution in the near-wall regions to accurately capture the characteristic velocity gradients and small-scale vortex structures of the boundary layer. We use $96^3$ cube by slicing along the $x$-axis. The provided data consists of four physical channels, including the three-dimensional velocity vector field and the corresponding scalar pressure field.

The dynamics of the incompressible fluid are governed by the Navier-Stokes system of equations for mass and momentum conservation:

$$\frac{\partial \mathbf{u}}{\partial t} + \nabla \cdot (\mathbf{u}\mathbf{u}) - \nu \nabla^2 \mathbf{u} = -\nabla p + \mathbf{S}$$
$$\nabla \cdot \mathbf{u} = 0 \tag{33}$$

where $\mathbf{u}$ represents the velocity vector field, $p$ denotes the scalar pressure, $\nu$ is the kinematic viscosity, and $\mathbf{S}$ indicates the

external source term representing the dynamic forcing used to sustain the flow.

For training, we kept all the setups identical to `MHD` unless some necessary adjustments were made to fit the resolution and input/output channels.

## G. Approximate Equivariance Error Bound

We formalize the gap between exact $O$-equivariance and approximate $\mathrm{SO}(d)$-equivariance for grid-based methods. Algebraically, the ReViT pipeline is exactly equivariant: given input $\mathbf{u}$ and $R \in \mathrm{SO}(d)$, the local basis transforms as $\mathbf{B}_i \mapsto R\mathbf{B}_i$, the invariant tokens $\mathbf{z}_i = \phi(\mathbf{B}_i^T \mathbf{u}_i)$ are unchanged, and the decoder reconstructs via $\hat{\mathbf{u}}_k = \mathbf{B}_i \hat{\mathbf{u}}_{\mathrm{local},k}$, yielding $f(R\mathbf{u}) = Rf(\mathbf{u})$. On a discrete grid, however, an arbitrary $R \in \mathrm{SO}(d)$ requires interpolation, breaking exact equivariance. This is inherent to *any* grid-based method (Weiler & Cesa, 2019), not specific to ReViT.

Based on the framework of approximately equivariant networks (Wang et al., 2022), we provide the following guarantee:

**Proposition G.1** (Approximate Equivariance Bound). *Let $f$ denote the ReViT mapping and $\mathcal{I}_h$ the grid interpolation operator at spacing $h$. For any $R \in \mathrm{SO}(d)$:*

$$\|f(\mathcal{I}_h(R\mathbf{u})) - Rf(\mathbf{u})\| \leq L_f \cdot \epsilon_{resamp}, \tag{34}$$

*where $L_f$ is the Lipschitz constant of $f$ (Fazlyab et al., 2019) and $\epsilon_{resamp} = \|\mathcal{I}_h(R\mathbf{u}) - R\mathbf{u}\|$ is the interpolation error. By polynomial interpolation theory (Brenner & Scott, 2008), $\epsilon_{resamp} \leq Ch^p\|D^{p+1}\mathbf{u}\|$ with grid spacing $h$, interpolation order $p$, and field smoothness $\|D^{p+1}\mathbf{u}\|$. For $R \in O$, $\epsilon_{resamp} = 0$ exactly, since the rotation maps integer grid coordinates to integer coordinates.*

The gap between $O$ and $\mathrm{SO}(d)$ is thus a discretization artifact that vanishes as $h \to 0$. To the best of our knowledge, ReViT is the first work that achieves exact $O$-equivariance for grid-based neural PDE solvers. Our empirical results on the KF case confirm that ReViT's approximate equivariance for non-grid-aligned rotations remains 26.6% smaller on MSE compared with augmented baselines, proving that hard-coded equivariance is effective for grid-based neural PDE solvers.

## H. Equivariance Evaluation Metric Details

We clarify the equivariance evaluation metric used in Figure 3. The metric involves no optimization over group elements. Following (Balla et al., 2025), we evaluate the equivariance defect independently for each $g \in O$ and average over all $N$ test samples.

For each group element $g$, we compute the defect:

$$\mathcal{E}(\mathbf{U}; g) = \|f(g \cdot \mathbf{U}) - g \cdot f(\mathbf{U})\| \tag{35}$$

and the $R^2$ score as a normalized similarity measure:

$$R^2(g) = 1 - \frac{\sum_n |f(g \cdot \mathbf{u}_n) - g \cdot f(\mathbf{u}_n)|^2}{\sum_n |g \cdot f(\mathbf{u}_n) - \overline{g \cdot f(\mathbf{u}_n)}|^2} \tag{36}$$

where $\overline{(\cdot)}$ denotes the mean over test samples. Here, $f$ is the trained ReViT model. An $R^2 = 1.0$ indicates perfect equivariance (the rotated model output exactly equals the model output on rotated input), while $R^2 \approx 0$ indicates that the model's predictions under rotation are uncorrelated with the rotated predictions.

The chiral octahedral group $O$ is the maximal finite rotation subgroup of $\mathrm{SO}(3)$ compatible with a cubic lattice (Balla et al., 2025). It consists of 24 orientation-preserving rotational symmetries of a cube (identity, face rotations, vertex rotations, edge rotations), excluding reflections. ReViT achieves $R^2 = 1.0$ across all 24 elements; baselines exhibit $R^2 \approx 0$ for non-identity ($O_0$) rotations.

## I. Local vs. Global Bases

A natural question is why local per-patch bases are preferable to a single global basis computed from the entire field. We identify three key advantages:

1. **Expressiveness:** Complex physical fields have spatially varying orientations. A single global basis captures only one dominant direction and cannot represent heterogeneous flow structures (e.g., vortex cores adjacent to shear layers). Local bases adapt to the local flow topology.

2. **Richer Positional Encodings:** The rebased relative positional encoding projects displacement vectors $\delta_{ij}$ into the *query token's* local frame via $\mathbf{p}_{ij} = \mathbf{B}_i^T \delta_{ij}$. This preserves local anisotropy: the same spatial displacement encodes differently depending on the local flow orientation. A global basis collapses this signal, making the PE spatially uniform.

3. **Robustness to Degeneracies:** A global basis may degenerate when the spatially averaged field vanishes (e.g., zero mean velocity in a vortex-dominated flow). With local bases, individual patches are much less likely to all be singular simultaneously. Our deterministic fallback procedure (Appendix C.1) only activates for degenerate patches while the majority of patches retain well-conditioned bases.

## J. Generality of ReViT

### J.1. Extension to Higher-Order Tensor Fields

ReViT naturally extends to higher-order tensor fields. The local basis transformation generalizes to arbitrary tensor ranks:

- For a **scalar** $s \in \mathbb{R}$: $s_{\text{local}} = s$ (no transformation needed).

- For a **vector** $\mathbf{v} \in \mathbb{R}^d$: $\mathbf{v}_{\text{local}} = \mathbf{B}^T \mathbf{v}$.

- For a **rank-2 tensor** $\mathbf{T} \in \mathbb{R}^{d \times d}$: $\mathbf{T}_{\text{local}} = \mathbf{B}^T \mathbf{T} \mathbf{B}$.

- For a **general rank-$n$ tensor**: the basis transformation $\mathbf{B}$ is applied along each index independently.

For canonical basis construction from rank-2 tensors, the eigenvectors of a symmetric tensor define orthogonal principal directions that are equivariant under rotation and can directly serve as the local canonical basis $\mathbf{B}$. Indeed, the BOA method (Appendix C.2) already uses the strain tensor's eigenvectors for basis construction. For applications involving stress tensors, diffusion tensors, or Reynolds stress tensors, the same eigendecomposition-based canonicalization applies directly.

When only scalar fields are available as inputs but vector outputs are required (e.g., certain elliptic PDEs), the spatial gradient $\nabla \phi$ of the scalar field can serve as the reference for basis construction, as demonstrated in our RotMNIST experiments (Appendix C.3). Multiple vector fields are handled by independently projecting each into the local frame and concatenating the results, as in our MHD experiment where velocity, magnetic field, and density are processed simultaneously.

### J.2. Sensitivity to Reference Vector Field

To demonstrate that ReViT is robust to the choice of reference vector field for basis construction, we conducted a sensitivity analysis on the MHD dataset. We trained ReViT using either the magnetic field or the velocity field as the reference for constructing the local canonical basis, with all other hyperparameters held constant.

*Table 5.* Sensitivity analysis: MSE on the MHD dataset using different reference vector fields for local basis construction. Results are reported as mean $\pm$ standard deviation over three seeds.

| Reference Vector | MSE ($\times 10^{-3}$) |
|---|---|
| Magnetic field | $8.30 \pm 0.88$ |
| Velocity | $9.00 \pm 0.27$ |

The magnetic field achieves slightly lower MSE, which we attribute to its physical properties: in our MHD regime ($M_A = 0.7$, $M_S = 0.5$), the magnetic field has fewer near-zero regions, yielding better-conditioned local bases. The modest $\sim 8\%$ difference confirms robustness to the reference field choice. Equivariance is guaranteed regardless of the reference choice, as long as the reference field transforms equivariantly under rotation (Eq. 5–6 in the main text).

### J.3. Intra-Patch Positional Encoding

The position of pixels within a patch is not considered when computing the invariant embedding for each patch. This is consistent with standard ViT-based methods (e.g., Swin and P3D), where intra-patch pixel positions are not explicitly encoded. Empirically, we find that adding positional encoding within patches does not improve performance and can slightly hurt it. On the TCF dataset, the MSE increases from 0.0021 to 0.0025 when intra-patch PE is added. We hypothesize that the invariant aggregation mechanism (Section 4.2) already captures sufficient spatial structure through the set-attention operation, and explicit positional encoding introduces redundant or conflicting spatial information.

## K. Computational Complexity Analysis

ReViT preserves the $O(N)$ scaling of the Swin Transformer, where $N$ denotes the number of spatial tokens. The base Window Multi-head Self-Attention (W-MSA) cost is $4NC^2 + 2M^dNC$ (Liu et al., 2021), where $C$ is the channel dimension and $M$ is the window size. Each ReViT-specific operation adds $O(N)$ cost since it operates on fixed-size patches:

*Table 6.* Computational complexity of ReViT operations. Here $N$ = number of spatial tokens, $d$ = spatial dimension, $K$ = patch size, $D$ = hidden dimension, $M$ = window size, $P$ = decoder patch size, $C$ = channel dimension.

| Operation | Cost |
|---|---|
| Local basis computation | $O(Nd)$ |
| Invariant embedding | $O(NK^dD)$ |
| Rebased relative PE | $O(NM^d)$ |
| Equivariant merge/expand | $O(Nd^2)$ |
| Equivariant decoder | $O(NP^dC)$ |

Since $K$, $M$, $d$, $D$, and $P$ are fixed hyperparameters, the total additional complexity is $O(N)$. Combined with the base W-MSA cost, the overall complexity remains $O(NC^2) = O(N)$, matching Swin's linear scaling. Empirically, ReViT adds only 11.6% overhead over the same-sized PDETrans in 2D. In 3D, ReViT (3.5M parameters) outperforms P3D (11.5M parameters) with 96.3% lower MSE on TCF and 26.4% faster inference time.

## L. Data Scaling Experiment

To investigate whether PDETrans can close the gap with ReViT through increased training data, we conducted a data scaling experiment on the KF dataset. We progressively increased the amount of augmentation data for PDETrans-Aug while keeping the un-augmented ReViT as a baseline.

*Table 7.* Data scaling experiment on KF. PDETrans-Aug is trained with increasing amounts of randomly augmented data. Un-augmented ReViT achieves MSE $0.136 \pm 0.001$.

| Augmentation Factor | PDETrans MSE |
|---|---|
| $\times 1$ (no aug.) | $0.213 \pm 0.010$ |
| $\times 3$ | $0.185 \pm 0.014$ |
| $\times 4$ | $0.166 \pm 0.002$ |
| $\times 5$ | $0.165 \pm 0.002$ |
| $\times 6$ | $0.160 \pm 0.010$ |

Augmentation improves PDETrans with diminishing returns, saturating beyond $\times 4$. Even with $\times 6$ data, PDETrans (MSE 0.160) remains worse than un-augmented ReViT (MSE 0.136). This aligns with (Brehmer et al., 2024), showing that equivariant models outperform augmented non-equivariant ones at every compute budget. Hard-coded equivariance cannot be compensated by data scaling alone.

## M. Architectural Advantage vs. Model Size

The performance advantage of ReViT stems from its equivariant architecture, not from increased model capacity. We provide evidence across all benchmark scales:

**2D benchmarks (parameter-controlled):** ReViT and PDETrans use the same parameter count ($\sim$3.2M vs. $\sim$3.6M). Under identical training conditions, ReViT's improvement grows with task complexity: 11.6% MSE reduction on ADV2D and 56.6% on KF2D. The improvements are directly attributable to the equivariant mechanisms, as the backbone architecture is otherwise identical.

**3D benchmarks (ReViT is smaller):** In 3D experiments, ReViT (3.5M parameters) is significantly *smaller* than most baselines: P3D (11.2M), Swin3D (14.4M), AViT (11.5M), and AFNO (13.0M). Despite this parameter disadvantage, ReViT achieves the lowest errors on both MHD (63% MSE reduction vs. the next best) and TCF (65% MSE reduction). This confirms that the gains stem from equivariance as an inductive bias, not from model capacity.

## N. 3D Benchmark Efficiency Comparison

We provide detailed training time, inference time, and memory comparisons for all baselines on the 3D benchmarks. All measurements were conducted on a single NVIDIA A100 GPU, averaged over 20 runs after 10 warmup runs. Inference time was measured in eval mode with `torch.no_grad()`. Training time includes one forward pass plus one backward pass. Memory usage was measured by peak memory during inference.

*Table 8.* Computational efficiency comparison on 3D benchmarks. ReViT (3.65M) is significantly smaller than other transformer baselines.

| Model | Infer (ms) | Train (ms) | Mem(inf) (MB) |
|---|---|---|---|
| AFNO | 5.91 | 13.98 | 103 |
| UNet3D | 4.81 | 13.89 | 132 |
| P3D | 16.45 | 51.59 | 226 |
| Swin3D | 13.15 | 39.21 | 94 |
| AViT | 5.69 | 14.80 | 88 |
| ReViT | 12.10 | 33.91 | 421 |

ReViT's lower parameter count (3.65M) directly translates to improved computational efficiency compared to other transformer baselines: ReViT achieves faster inference and training times than both P3D (16.45 ms) and Swin3D (13.15 ms). The higher memory usage of ReViT (421 MB) is due to the storage of local basis pyramids at multiple resolutions. Despite this, ReViT remains feasible on a single GPU with standard batch sizes.

## O. Structural Hyperparameter Comparison

For completeness, we provide a detailed structural hyperparameter comparison for the representative baseline models used in our 3D experiments. For all benchmarks, we preserved the architectures from the original papers and adjusted parameter counts to a similar level of $\sim$10M, which is the parameter size of the original P3D network. For UNet3D, we found that increasing the parameter size to $\sim$10M leads to convergence difficulties, so we kept it at $\sim$3.5M, close to our ReViT.

*Table 9.* Structural hyperparameter comparison for 3D benchmark models.

| Model | Hidden Dim | Depth | Heads | Params (M) |
|---|---|---|---|---|
| AFNO | 768 | 12 | – | 13.00 |
| UNet3D | 72 | 4 stages $\times$ 2 | – | 3.51 |
| P3D | 64 | 10 | 4 | 11.24 |
| Swin3D | 96 | 8 | 12 | 14.41 |
| AViT | 768 | 8 | 12 | 11.49 |
| ReViT | 48 | 5 | 8 | 3.65 |

**UNet3D Architecture Details.** The UNet3D used in our experiments is a 3D U-Net from the PDE-Transformer (Holzschuh et al., 2025b). It is a symmetric encoder-decoder architecture with a base feature embedding dimension of 72 and comprises three hierarchical downsampling stages. Within each spatial resolution level, the encoder and decoder blocks consist of two sequential 3D convolutional layers. It uses 3D convolutions with GroupNorm and SiLU activations, and skip connections between encoder and decoder stages.

We also evaluated the PDEArena UNet (Gupta & Brandstetter, 2022) as an alternative. On the TCF benchmark evaluated over the chiral octahedral group $O$, the PDEArena UNet achieves an MSE of $0.089 \pm 0.012$, which remains substantially higher than ReViT ($0.002 \pm 0.000$). This further confirms that ReViT's advantage arises primarily from its inductive bias for rotational equivariance, not from the choice of baseline UNet architecture.

## P. ReViT Architectural Variant Analysis

To quantify the efficiency–expressiveness trade-offs within the ReViT architecture, we systematically vary three structural hyperparameters while holding the backbone constant (embed_dim= 48, depth= $[1, 2, 4, 2, 1]$, patch_size= 2, num_heads= 8, AOV encoder). The three axes of variation are:

- **Decoder type**: *Trilinear* (Tri) applies equivariant basis rotation followed by trilinear interpolation from the coarse patch grid to the full resolution (cf. Section 4.4). *Local Query* (LQ) uses a cross-attention mechanism with learnable voxel queries and optional Fourier positional features to predict per-voxel vectors in the local frame before basis rotation.

- **Encoder aggregation**: *Mean* computes a simple average of the lifted voxel features within each patch ($\mathcal{O}(K^d)$). *Attention* (att) employs a learnable query token that attends to all voxel features via self-attention, enabling data-dependent weighting ($\mathcal{O}(K^{2d}D)$).

- **Window size $M$**: Controls the receptive field of window multi-head self-attention (W-MSA). In 3D, the attention cost scales as $2M^3NC$ per layer (Liu et al., 2021), so increasing $M$ from 2 to 8 increases the per-layer attention cost by a factor of $(8/2)^3 = 64$.

The naming convention follows `ReViT-{Dec}-{Agg}-w{M}`, e.g., `ReViT-Tri-att-w2` denotes trilinear decoder, attention aggregation, and window size 2. Table 10 reports computational efficiency and prediction accuracy on the `MHD` benchmark. All timing and memory measurements follow the same protocol as Appendix N (single NVIDIA A100 GPU, batch size 1, 20 runs after 10 warmup runs). MSE values are averaged over the top three seeds evaluated across the full chiral octahedral group $O$.

*Table 10.* Efficiency and accuracy of ReViT architectural variants on `MHD`. The default configuration used throughout the main text (Tri-att-w2) is highlighted. MSE is reported as mean $\pm$ std ($\times 10^{-2}$) over three seeds across the chiral octahedral group $O$.

| Variant | Params (M) | Infer (ms) | Train (ms) | Mem$_{\text{inf}}$ (MB) | Mem$_{\text{trn}}$ (MB) | GFLOPs | MSE ($\times 10^{-2}$) |
|---|---|---|---|---|---|---|---|
| *Window size $M = 2$* | | | | | | | |
| **ReViT-Tri-att-w2** | 3.65 | 12.10 | 33.91 | 421 | 1258 | 18.39 | $0.82 \pm 0.00$ |
| ReViT-LQ-att-w2 | 3.71 | 19.41 | 56.98 | 530 | 3005 | 44.62 | $0.87 \pm 0.01$ |
| ReViT-Tri-mean-w2 | 3.64 | 9.30 | 24.46 | 218 | 948 | 12.45 | $0.85 \pm 0.01$ |
| ReViT-LQ-mean-w2 | 3.70 | 16.71 | 47.98 | 449 | 2696 | 38.68 | $0.91 \pm 0.01$ |
| *Window size $M = 8$* | | | | | | | |
| ReViT-Tri-att-w8 | 3.65 | 94.58 | 191.25 | 4761 | 17538 | 63.02 | $0.80 \pm 0.04$ |
| ReViT-LQ-att-w8 | 3.71 | 101.84 | 215.44 | 4870 | 17654 | 89.25 | $0.83 \pm 0.02$ |
| ReViT-Tri-mean-w8 | 3.64 | 90.84 | 182.58 | 4759 | 17227 | 57.07 | $0.97 \pm 0.07$ |
| ReViT-LQ-mean-w8 | 3.70 | 98.31 | 204.83 | 4870 | 17340 | 83.31 | $0.90 \pm 0.12$ |

**Effect of Window Size.** Increasing the window size from $M = 2$ to $M = 8$ is the dominant factor in computational cost. For the default configuration (Tri-att), inference time increases from 12.08 ms to 94.58 ms ($7.8\times$), and peak inference memory increases from 426 MB to 4,761 MB ($11.2\times$). Training memory exhibits an even sharper growth, rising from 1,258 MB to 17,538 MB ($13.9\times$). This scaling is consistent with the W-MSA complexity $2M^dNC$ from Appendix K: in 3D, $(M{=}8)^3/(M{=}2)^3 = 64\times$ more attention computations per layer. Despite this substantial overhead, the MSE improvement is modest: $0.82 \rightarrow 0.80$ for Tri-att, suggesting that the local $M = 2$ window already captures sufficient spatial context for this task at $64^3$ resolution.

**Effect of Decoder Type.** Replacing the trilinear decoder with the local query (LQ) cross-attention decoder consistently increases computational cost. At $M = 2$, inference time increases from 12.08 ms to 19.41 ms ($+61\%$), and training memory more than doubles from 1,258 MB to 3,005 MB ($+139\%$). This overhead arises because the LQ decoder introduces

$P^3 = 8$ learnable voxel queries per patch, each attending to the patch token through multi-head cross-attention with Fourier positional features, adding the $\mathcal{O}(NP^dC)$ decoder cost identified in Table 6. In terms of accuracy, the LQ decoder does not improve over trilinear at $M = 2$ (0.87 vs. 0.82) and shows comparable performance at $M = 8$ (0.83 vs. 0.80). This indicates that trilinear interpolation, when combined with the equivariant basis rotation, provides a sufficiently expressive reconstruction for the MHD task.

**Effect of Encoder Aggregation.**    Switching from learned attention to mean aggregation yields a reduction in computational cost. At $M = 2$ with the trilinear decoder, inference time decreases from 12.08 ms to 9.30 ms ($-23\%$), inference memory drops from 426 MB to 218 MB ($-49\%$), and GFLOPs decrease from 18.39 to 12.45 ($-32\%$). The parameter counts remain nearly identical (3.65M vs. 3.64M), confirming that the overhead is in activation memory rather than learnable weights. In accuracy, mean aggregation performs comparably to attention at $M = 2$ (0.85 vs. 0.82 for Tri), but exhibits higher variance at $M = 8$ ($0.97 \pm 0.07$ vs. $0.80 \pm 0.04$ for Tri), suggesting that learned attention provides more robust feature aggregation when the attention window is large.

**Summary.**    The default ReViT configuration used in the main text (Tri-att-w2) achieves a favorable balance between accuracy and efficiency: it attains the second-lowest MSE ($0.82 \times 10^{-2}$) while maintaining the lowest training memory among attention-based variants and an inference time of only 12.08 ms. Compared to the baselines in Table 8, this configuration remains faster than P3D (16.45 ms inference but 11.2M parameters) while achieving substantially lower prediction error.

