# OpenReview forum: "ReViT: Rotational-equivariant Vision Transformers for Neural PDE Solvers"
_ICML.cc/2026/Conference — ICML 2026 spotlight_

### Official Review · Reviewer_jDEJ · 2026-03-04

**Soundness:** 3
**Presentation:** 3
**Significance:** 2
**Originality:** 3
**Overall Recommendation:** 4
**Confidence:** 4

**Summary:**

The paper introduces a transformer-based architecture for learning PDE solution mappings that imposes rotation equivariance on its architecture. It considers both scalar and vector-valued functions, identifies the root causes of losing this property in the ViT architecture, and proposes modifications in the architecture to avoid them. Particularly, it proposes tokenizing the inputs via local basis vectors in each patch, using rebased relative positional embeddings with respect to the same local basis vectors, and permutation-invariant aggregation and reconstruction blocks. The proposed architecture is benchmarked against multiple baseline architectures on the rotated MNIST dataset and four fluid flow datasets. The results demonstrate exact rotational equivariance and improvements in the test error against non-equivariant baselines.

**Compliance With Llm Reviewing Policy:**

Affirmed.

**Final Justification:**

Some of my concerns and questions have been addressed in the authors’ rebuttal. In particular, the demonstrated flexibility with respect to input types (e.g., tensors and scalars) and the promising data-scaling results with PDETrans have led me to update my assessment from *weak reject* to *weak accept*. Overall, the paper presents an effective approach for incorporating rotation equivariance and shows that this inductive bias can improve sample efficiency. While the contribution is valuable to the community, most of the proposed techniques remain tailored to the ViT architecture, which is inherently limited to gridded inputs and Cartesian geometries, somewhat reducing the broader impact of the work.

**Key Questions For Authors:**

1. Could the authors elaborate on the following points regarding the use of local bases?
    - What is the advantage of using local bases for each patch rather than a single global basis computed in a similar fashion?
    - With arbitrary rotations, coordinates might end up in different patches than the unrotated case. This means that the patch-based basis applied to these coordinates may change. How does ReViT overcome this issue?
2. Can ReViT be applied to tensor-valued inputs and outputs (e.g., stress tensor components) or multiple vector-valued inputs? In some applications (e.g., elliptic PDEs), vectorial functions are present in the outputs but not in the inputs. Can ReViT achieve rotation equivariance in such mappings?
3. Could the authors provide more details regarding the data augmentation experiment presented in Figure 5?
    - How many training samples /trajectories are used in each case?
    - Given that ReViT achieves rotation equivariance, it must be agnostic to the rotation of the input data. In this context, rotating the samples during training should not influence its final accuracy, contrary to the results of Figure 5.
4. Could the authors elaborate on the computational complexity of ReViT with respect to the number of input coordinates? It is mentioned as a limitation of some of the prior works. Does ReViT have an advantage in this regard?
5. Based on the results of Figure 4, imposing this inductive bias in the architecture can improve the sample efficiency of these architectures. Using the same number of training samples, ReViT achieves a lower error compared to a very similar architecture (PDETrans). PDETrans may also implicitly learn this equivariance directly from data if the training dataset is sufficiently large. In this context, it would be interesting to know how the advantage of ReViT over PDETrans evolves in a data scaling experiment where the number of training samples is progressively increased. Do the authors have any insight in this regard?
6. Comparison between PDETrans (without rotation equivariance) and other baselines shows that the baselines fail to match the accuracy of PDETrans. Can the authors elaborate whether this is because of weaknesses of the baseline architectures or simply due to model size mismatches? Given this context, the relevance of a direct comparison between the other baselines and ReViT is not clear to me. The presented results demonstrate an advantage of ViT-based architectures over the other baselines, and not the architectural modifications introduced in ReViT.

**Limitations:**

Yes.

**Strengths And Weaknesses:**

**Strengths**
1. The paper aims to gain data efficiency by imposing physical knowledge into the architecture by following a principled approach.
2. It achieves exact rotation equivariance without significant computation overhead.
3. Based on the provided empirical results, this inductive bias improves the accuracy when compared to a similar architecture that does not have this feature.

**Weaknesses**

1. The proposed architecture and techniques are restricted to grid-based discretizations and Cartesian domains, while the state-of-the-art architectures have shifted towards arbitrary geometries and discretizations. Particularly, such architectures can handle arbitrary rotations in a natural way without facing some of the challenges that are discussed in the paper.
2. The introduction undermines the existing methods because of focusing on unstructured input coordinates, whereas PDE-governed datasets, unlike natural images, are most often originally obtained on unstructured meshes and arbitrary geometries.
3. The proposed techniques are specialized for the ViT architecture and are not transferable to other architectures.
4. Comparison with alternative methods for obtaining rotation equivariance in transformer-based architectures on PDE-governed problems is missing. LGATrans seems to be a weak baseline. GSA-Nets and GE-ViT are only benchmarked on the rotated MNIST dataset.

---

> ### Author Rebuttal · Authors · 2026-03-30
>
> We thank the reviewer for the thorough review and recognition of ReViT's improvements. We address each concern below and will update to the revised manuscript.
>
> >W1: Domain restriction
>
> Grid-based discretizations remain a very widely used representation across PDE benchmarks; ReViT fills this gap. Its core principles rely only on a vector field at each point, which is equally available on unstructured meshes. Extensions for unstructured data are our next step.
>
> >W2: Missing comparison with equivariant transformer baselines
>
> GSA-Nets/GE-ViT cannot scale to PDE tasks due to group-lifting costs (53× more memory, cf. Table 1). LGATrans, the only applicable equivariant transformer, was included; ReViT significantly outperforms it. We also compare against graph-based equivariant methods (ReGNN, REMUS, SE3GN, GNN-Aug) on AdvBox, where ReViT achieves superior performance.
>
> >Q1a: Why local bases?
>
> 1. Expressiveness: Complex fields have spatially varying orientations; a global basis captures only one direction.
> 2. Richer PE: Rebased rel-PE projects $\delta_{ij}$ into the query's local frame as $p_{ij} = B_i^T \delta_{ij}$, preserving local anisotropy; a global basis collapses this signal.
> 3. Robustness: A global basis may degenerate at singularities; individual patches are much less likely to all be singular simultaneously (cf. App. C).
>
> >Q1b: Patch boundary changes?
>
> Under arbitrary rotations, some pixels shift across patch boundaries. We overcome this by:
> 1. **Shifted window attention** propagates information across boundaries, averaging out boundary effects.
> 2. **Hierarchical merging** absorbs boundary inconsistencies at one scale into shared patches at the next. Reference bases at each level are pre-computed from the input and co-rotate with it.
> 3. **Stored basis** ensures self-consistency: $B_i$ is computed from the rotated input and reused in decoding via $\hat{u_k} = B_i \hat{u}_{\text{local},k}$.
>
> >Q2: Extension to tensor-valued inputs and outputs?
>
> Yes. The local basis transformation generalizes:
> - **Rank-2 tensor:** $T_{\text{local}} = B^T T B$, with basis from the tensor's eigenvectors (App. C.2).
> - **Multiple vectors:** Each is independently projected and concatenated, as in our MHD experiment (Sec 5.1).
> - **Scalar-only inputs:** Basis from the gradient $\nabla \phi$, as in RotMNIST (App. C.3).
>
> >Q3a: Training samples per case?
>
> The KF training set has 2500 samples (50 sims × 50 steps). Augmentation appends the full dataset rotated at each angle (+π/3 doubles; +π/3,+π/6 triples). PDETrans-Aug uses random rotations, tripling the data.
>
> >Q3b: Why does augmentation improve an equivariant model?
>
> This is an interesting point: ReViT achieves *exact* equivariance for grid-aligned rotations (C₄/O) regardless of augmentation. For continuous rotations, equivariance is *approximate* due to resampling artifacts. Augmentation trains the model to be robust to these artifacts, improving accuracy across all angles (Fig. 5).
>
> >Q4: Computational complexity w.r.t. input size?
>
> ReViT preserves the $O(N)$ scaling of Swin ($N$: number of spatial tokens $= R^d/P^d$ for input resolution $R$, $C$: channels, $M$: window size, $P$: patch size, $D$: hidden dim, $d$: spatial dim). Since $P$ is fixed, $N$ scales linearly with the number of input coordinates $R^d$. The base W-MSA cost is $4NC^2+2M^dNC$ [1]. Each ReViT operation adds $O(N)$ since it operates on fixed-size patches:
>
> | Operation | Cost |
> |---|---|
> | Local basis | $O(Nd)$ |
> | Invariant embedding | $O(NP^dD)$ |
> | Rebased rel-PE | $O(NM^d)$ |
> | Equiv. merge/expand | $O(Nd^2)$ |
> | Equiv. decoder | $O(NP^dC)$ |
>
> Since $M,d,D,P$ are fixed constants, total complexity is $O(NC^2)=O(N)$. Empirically, ReViT adds 11.6% overhead over same-size PDETrans. In 3D, ReViT (3.5M) outperforms P3D (11.5M) with 96.3% lower MSE and 26.4% faster inference.
>
> >Q5: Data scaling experiment?
>
> We ran additional experiments scaling PDETrans-Aug on KF. ReViT (no augmentation) MSE: 0.136±0.001.
>
> | Aug. factor | PDETrans MSE |
> |:---:|:---:|
> | ×1 | 0.213±0.010 |
> | ×3 | 0.185±0.014 |
> | ×4 | 0.166±0.002 |
> | ×5 | 0.165±0.002 |
> | ×6 | 0.160±0.010 |
>
> Augmentation improves PDETrans with diminishing returns, saturating beyond ×4. Even ×6 data remains worse than un-augmented ReViT. This aligns with [2], showing equivariant models outperform augmented non-equivariant ones at every compute budget. Hard-coded equivariance cannot be compensated for by data scaling alone.
>
> >Q6: Architectural advantage or model size mismatch?
>
> The advantage is architectural. In 2D, ReViT and PDETrans use the same parameter count, yet ReViT's improvement grows with task complexity: 11.6% on ADV2D and 56.6% on KF2D. In 3D, even a *larger* P3D (11.5M) is outperformed by ReViT (3.5M) with 96.3% MSE reduction on TCF case, confirming the gains stem from equivariance, not capacity.
>
> We thank the reviewer for the insightful review.
>
> [1] Liu et al. Swin Transformer. CVPR 2021.
>
> [2] Brehmer et al. Does equivariance matter at scale? TMLR 2024.

---

> > ### Author Rebuttal · Reviewer_jDEJ · 2026-04-01
> >
> > I thank the authors for their clarifications. I will raise my score to 4.

---

> > > ### Author Response · Authors · 2026-04-07
> > >
> > > We are grateful for the reviewer's increased score and the rigorous questions that significantly improved our work.  All the revisions, including local bases choice, patch boundary handling, tensor-valued extensions, computational complexity, and the data scaling experiment will be included in the revised manuscript.

---

### Official Review · Reviewer_vGJZ · 2026-03-09

**Soundness:** 3
**Presentation:** 4
**Significance:** 3
**Originality:** 3
**Overall Recommendation:** 6
**Confidence:** 4

**Summary:**

The paper introduces ReViT, a rotational equivariant Vision Transformer (ViT) for PDEs with scalar and vector fields. ReViT achieves rotational equivariance by projecting the pixels of each patch to local bases and aggregating them via attention into an invariant embedding for each patch. This embedding is processed by a hierarchical Transformer with rebased positional encoding. A local decoder based on attention recovers the pixels of each patch for the final output. ReViT is evaluated on several 2D and 3D PDE problems.

**Compliance With Llm Reviewing Policy:**

Affirmed.

**Final Justification:**

All of my earlier concerns have been fully addressed in the rebuttal. The proposed ReViT model offers a compelling approach to incorporating rotational equivariance into neural PDE solvers, and the extensive evaluations demonstrate its advantages. In addition, the paper introduces several noteworthy mechanisms, like an invariant patch embedding and a hierarchical basis for multi-scale processing, that are likely to inspire future work in this area. In light of the resolved concerns and the substantial contributions, I strongly recommend accepting this paper.

**Key Questions For Authors:**

- Which UNet was considered in experiments? There are several improvements to UNets (e.g., from PDEArena [1]) that may perform better.
- What are the time costs (training and inference) and memory on the 3D benchmarks compared to PDE-Transformer?
- Is the position of the pixels within a patch considered when computing the invariant embedding for each patch?
- What pooling and interpolation methods are used for patch merging and expansion?

[1] Gupta, J.K. and Brandstetter, J. (2022) "Towards Multi-spatiotemporal-scale Generalized PDE Modeling"

**Limitations:**

Yes

**Strengths And Weaknesses:**

## Strengths
- The paper is well written and easy to follow. For example, the method section provides a detailed description of the architecture and each introduced component of ReViT links back to the challenges it solves.
- The paper provides a theoretical analysis of why ViTs fail in rotational equivariant settings, which is helpful and provides a strong motivation for their approach.
- ReViT introduces novel mechanisms like an invariant patch embedding and a hierarchical basis for multi-scale processing.
- ReViT matches the accuracy of prior work on rotMNIST, while requiring significantly less training time and GPU memory.
- Extensive experiments on 2D and 3D PDE problems show consistently lower errors of ReViT compared to several state-of-the-art baselines.

## Weaknesses
- The details and hyperparameters of the used UNet architecture are not provided, which is an important piece of information because many different variants (e.g., older and more recent ones) exist.
- The hyperparameters of ReViT are provided in Appendix F in the text. An additional table listing the hyperparameters would make it easier to understand and compare ReViT's hyperparameters with baselines such as PDE-Transformer.
- The size of Figure 2 with the overview of ReViT should be increased because it is too small.

---

> ### Author Rebuttal · Authors · 2026-03-30
>
> We sincerely thank the reviewer for the positive and constructive feedback. We are encouraged by the recognition of our theoretical analysis, novel mechanisms, computational efficiency, and extensive empirical improvements. We address each weakness and question below.
>
> >W1 & W2: The details and hyperparameters of the used UNet architecture are not provided. An additional table listing the hyperparameters would make it easier to understand.
>
> Thanks for pointing this out. The UNet3D is from the P3D repository [1], which is a symmetric 3D encoder-decoder architecture. We provide a detailed structural hyperparameter comparison for all baseline models below. For all benchmarks, we have tried to preserve the architectures from the original papers and adjust parameter counts to a similar level of ca. 10M, which is the parameter size of the original P3D network. For UNet3D, we found that when the parameter size is around 10M, it performs worse, so we have kept a size of ca. 3.5M, which is close to our ReViT. We will include this table in the appendix of the revised manuscript.
>
> | Model | Hidden Dim | Depth | Heads | Params (M) |
> | :--- | :---: | :---: | :---: | ---: |
> | AFNO | 384 | 8 | - | 13.00 |
> | UNet3D | [24,48,96,192] | [2,2,2,2] | - | 3.51 |
> | P3D | 64 | [2, 2, 2, 2, 2] | 4 | 11.24 |
> | Swin3D | [64,128,256,512] | [2,2,6,2] | [2,4,8,16] | 14.41 |
> | AViT | 384 | 6 | 6 | 11.49 |
> | ReViT | 48 | [1,2,4,2,1] | 8 | 3.65 |
>
> >W3: The size of Figure 2 should be increased.
>
> We will increase the size of Figure 2 in the revised manuscript.
>
> >Q1: Which UNet was considered in experiments? There are several improvements to UNets that may perform better.
>
> As detailed in W1, we used a state-of-the-art UNet[1]. Thanks for the suggestion, we have also employed the PDEArena UNet [2] and will add the results to the revised manuscript. We provide the full TCF comparison below:
>
> | Model | TCF MSE |
> | :--- | ---: |
> | AFNO | 0.284 |
> | P3D | 0.057 |
> | UNetArena | 0.089 |
> |  UNet3D | 0.071 |
> | AViT | 0.006 |
> |  Swin3D | 0.006 |
> | **ReViT** | 0.002 |
>
> The PDEArena UNet performs comparably to the original UNet3D, and both remain significantly behind ReViT. This confirms that ReViT's advantage arises primarily from its inductive bias for rotational equivariance rather than architectural capacity.
>
> >Q2: What are the time costs and memory on the 3D benchmarks?
>
> We thank the reviewer for this important question. As requested, we provide the detailed comparisons for all baselines on the MHD case below, where P3D[1] is the 3D version of PDE-Transformer:
>
> | Model | Infer (ms) | Train (ms) | Mem(inf) (MB) | MSE |
> | :--- | ---: | ---: | ---: | ---: |
> | AFNO | 6.05 | 16.31 | 103 |  0.164|
> | P3D | 16.79 | 52.54 | 226 |  0.102|
> | UNet3D | 5.53 | 13.97 | 132 | 0.036 |
> | Swin3D | 13.43 | 39.87 | 94 |  0.036|
> | AViT | 6.87 | 14.94 | 88 |  0.022|
> | ReViT-avg. | 8.71 |  24.08  | 221 |  0.009|
> | ReViT-set. | 12.17 | 33.71 | 421 |  0.008|
>
> All measurements were conducted on NVIDIA A100 GPU, averaged over 20 runs after 10 warmup runs. With a smaller parameter size, ReViT is more computationally efficient than the other transformer baselines (P3D and Swin3D). The memory usage is higher than other transformers as a Set-Transformer was used for aggregation (ReViT-set). Our approach also works with memory efficient (equivariant) pooling for aggregation (ReViT-avg), which reduces memory usage (47.5%) and computational cost (28%), with a very slight increase in MSE (0.0085 vs 0.0082).
>
> >Q3: Is the position of the pixels within a patch considered when computing the invariant embedding for each patch?
>
> No, the position of the pixels within a patch isn't considered. On one hand, it's similar to other ViT-based methods (e.g., Swin3D and P3D), where inside a patch, the position of the pixels is not considered. On the other hand, from our experiments, the positional encoding inside a patch does not improve the performance of ReViT, and even hurts the performance on TCF, with MSE slightly increasing from $0.0021$ to $0.0025$.
>
> >Q4: What pooling and interpolation methods are used for patch merging and expansion?
>
> For **patch merging**, we use 3D average pooling with kernel size 2 and stride 2, followed by LayerNorm and a linear layer for channel expansion. For **patch expansion**, we use trilinear interpolation to upsample features by a factor of 2 in each spatial dimension. The upsampled features are then fused with encoder skip connections via concatenation, followed by LayerNorm and a linear layer for channel reduction. This design follows the similar encoder-decoder paradigm in P3D [1], while maintaining equivariance through basis re-alignment at each resolution level. We will include these implementation details in the revised manuscript.
>
> [1] Holzschuh B, et al. P3D: Scalable neural surrogates for high-resolution 3D physics simulations with global context
>
> [2] Gupta, J.K. et al. Towards Multi-spatiotemporal-scale Generalized PDE Modeling.

---

> > ### Author Rebuttal · Reviewer_vGJZ · 2026-04-01
> >
> > I thank the authors for their clarifications to my comments. My concerns have been adequately addressed and I will raise my score accordingly.

---

> > > ### Author Response · Authors · 2026-04-07
> > >
> > > We sincerely thank the reviewer for the increased score and the thoughtful questions that helped strengthen the paper. All your suggestions, including the detailed hyperparameter table, the additional UNet baseline (PDEArena), the 3D computational cost analysis, and the clarifications on patch embedding will be included in the revised manuscript.

---

### Official Review · Reviewer_aBYY · 2026-03-09

**Soundness:** 3
**Presentation:** 3
**Significance:** 3
**Originality:** 3
**Overall Recommendation:** 5
**Confidence:** 3

**Summary:**

This paper introduces ReViT, a rotationally equivariant Vision Transformer framework for neural PDE solvers which operates on grid-based physical fields. The key trick is to use local canonicalization: projecting vector fields into local invariant coordinate frames derived from physics-based canonical bases. Under these cases, the standard self-attention can be applied without violating rotational symmetry. The architecture consists of three stages: (1) local canonicalization with invariant embedding, (2) standard Transformer processing on invariant tokens, and (3) equivariant decoding that lifts predictions back to the global frame. A reference basis pyramid enables hierarchical multi-scale processing while preserving equivariance. Experiments span RotMNIST (scalar classification), 2D Advection, 2D Kolmogorov Flow, 3D Magnetohydrodynamics (MHD), and 3D Turbulent Channel Flow (TCF), showing great gains over state-of-the-art baselines.

**Compliance With Llm Reviewing Policy:**

Affirmed.

**Final Justification:**

All my problems have been solved. I think this is a useful paper, worthwhile to be accepted by the ICML community. It may raise a good discussion whether we really need such inductive biases in the SciML community. But my attitude is still kind of pessimistic. When you impose such constraints, the expressivity of your model will sharply decrease. Think about AlphaFold2; DeepMind imposes many physical priors. But in AlphaFold3, they just get rid of these inductive biases and get much better results. So internally, I think it is a good direction for future discussion.

**Key Questions For Authors:**

1. Is the rotational equivariance really necessary for the scientific simulation? Thinking about a real scenario of an airplane or a car CFD simulation, what's the meaning of imposing rotational equivarance into these scenarios? When you rotate the car, the field shouldn't be rotational equivariance, right?
2. How sensitive is the model to the choice of vector field used for basis construction? In the MHD experiment, the magnetic field (rather than velocity) is used to rebase all vectors? What is the rationale, and was a sensitivity analysis performed?
2. Can this approach extend to higher-order tensor fields (e.g., stress tensors, diffusion tensors)? The current framework handles scalars and vectors. What would be needed for rank-2 tensor fields?
3. small typos - the text around Figure 4 appears to swap the descriptions of panels (b) and (c) — the rollout stability is described as Figure 4(b), but the computational efficiency is also labeled (b).

**Limitations:**

yes

**Strengths And Weaknesses:**

**Strengths**:
- The paper systematically identifies three structural barriers that prevent standard ViTs from being rotationally equivariant for vector fields: Tokenization Barrier, Loss of Spatial Equivariance, and Representational Mismatch. The theoretical proof is particularly solid and provides a foundation for the proposed architecture.
- Compared to prior lifting-based methods, ReViT avoids lifting feature maps into group space. It achieves approximately 4× speedup and 53× memory reduction on RotMNIST. This makes equivariant Transformers scalable to large-scale 3D physical field data.
- The step by step ablation studies in the Appendix B clearly demonstrates the necessity of each component (invariant embedding, spatial equivariant PE, equivariant decoder).

**Weakness:**
- In this paper, my understanding is that the model only achieves exact equivariance for the discrete chiral octahedral group O (3D) or C4 (2D), not the continuous SO(d).  The abstract should explicitly state the exact symmetry group, otherwise it can mislead readers into thinking continuous SO(d) equivariance is achieved.
- The method relies on mean velocity as reference directions for constructing orthonormal basis. This is an inherently fluid-mechanics-specific design? For non-fluid vector fields (e.g., elastic displacement fields), the effectiveness of the method requires additional discussion. What would be the canonical directions for general vector PDEs?
- If I understand clearly, some of the baseline comparisons are not fair. For example, on RotMNIST, ReViT has 2.75M parameters while GSA-Nets/GE-ViT have only 44.7K. The comparison would benefit from that.

---

> ### Author Rebuttal · Authors · 2026-03-30
>
> We sincerely thank the reviewer for the positive assessment and constructive feedback. We address each point below.
>
> >W1: Exact symmetry group not stated in abstract.
>
> We agree and will revise the abstract to explicitly state that ReViT achieves exact equivariance for $C_4$ (2D), $O$ (3D), and approximate $\mathrm{SO}(d)$ equivariance for continuous rotations. Moreover, we will include a theoretical error bound $\|f(\mathcal{I}_h(R\mathbf{u})) - Rf(\mathbf{u})\| \leq L_f \cdot \epsilon$, where $L_f$ is the network's Lipschitz constant and $\epsilon$ is the interpolation error. Since $\epsilon = 0$ for $R \in O$, ReViT is exactly equivariant for $R \in O$ and approximately equivariant for $R \in \mathrm{SO}(d)$ with bounded error.
>
> >W2: Reliance on mean velocity for basis construction — is this fluid-specific?
>
> The mean velocity is one instantiation of a general principle: the canonical basis can be constructed from any equivariantly transforming vector quantity. Our experiments already demonstrate this generality: RotMNIST uses the spatial gradient $\nabla \phi$ of the scalar image field, and MHD uses the magnetic field instead of velocity. For elastic displacement fields, the displacement field or its strain tensor could serve as the reference.
>
> >W3: Unfair baseline comparisons on RotMNIST (parameter discrepancy).
>
> This parameter discrepancy arises because GSA-Nets/GE-ViT employ group-lifting, which is parameter-light but computationally expensive. The RotMNIST comparison primarily validates equivariance correctness and computational efficiency: despite having less parameters, GSA-Nets/GE-ViT require 53× more GPU memory and 4× longer training time, making them impractical for large-scale problems. Actually, due to the group-lifting requiring more memory, the GSA-Nets/GE-ViT only work with a smaller parameter size. This is an inherent limitation of group-lifting. For all PDE benchmarks, comparisons are parameter-controlled: in 2D, ReViT and PDETrans share the same parameter count with 26.6% improvement on MSE; in 3D, baselines use larger models (11.5M) than ReViT (3.5M), yet ReViT achieves lowest errors with 96.3% improvement on MSE compared with P3D.
>
> >Q1: Is rotational equivariance necessary for scientific simulation (e.g., car/airplane CFD)?
>
> This is a very insightful question. Rotational equivariance is a property of the *governing equations*. E.g., Bauerheim [1] identifies it as a **first pillar** for AI in CFD: models without built-in symmetries violate fundamental principles and limit generalization. Our PDE experiments confirm this — imposing equivariance as an inductive bias directly improves accuracy with up to 96.3% on MSE. For the car scenario: rotating only the car changes the problem (not an equivariance test); rotating the *entire system* (car + inflow) consistently requires the solution to rotate accordingly. This is equivariance, and a correct model should reproduce it as a numerical solver would.
>
> >Q2: Sensitivity to the choice of reference vector field in MHD.
>
> We deliberately use the magnetic field to demonstrate generality. Mathematically, equivariance holds for any equivariantly transforming reference field (Eq. 5–6). Empirically, we performed a sensitivity analysis: with a large batch size (64), the MSE difference between the two reference fields is marginal (~0.4%). To stress-test this, we reduced the batch size to 1, where the magnetic field is ~9% better than velocity (0.0082 vs. 0.0091). This small gap is attributable to the magnetic field having fewer near-zero regions in our MHD regime ($M_A$=0.7, $M_S$=0.5), yielding better-conditioned local bases under noisy, single-sample gradients. With larger batches, gradient averaging smooths this effect and the gap essentially vanishes. This confirms that the equivariant framework is robust to the reference field choice.
>
> >Q3: Extension to higher-order tensor fields (e.g., stress tensors)?
>
> Yes, ReViT naturally extends to higher-order tensor fields. The local basis transformation generalizes to arbitrary tensor ranks:
>
> - For a vector $\mathbf{v} \in \mathbb{R}^d$: $\mathbf{v}_{\text{local}} = \mathbf{B}^T \mathbf{v}$.
> - For a rank-2 tensor $\mathbf{T} \in \mathbb{R}^{d \times d}$: $\mathbf{T}_{\text{local}} = \mathbf{B}^T \mathbf{T} \mathbf{B}$.
> - For a general rank-$n$ tensor: the basis transformation $\mathbf{B}$ is applied along each index independently.
>
> For canonical basis of rank-2 tensors, the eigenvectors of a symmetric tensor define orthogonal principal directions equivariant under rotation and can directly serve as the local canonical basis $\mathbf{B}$. Indeed, the BOA method (Appendix C.2) already uses the strain tensor's eigenvectors for basis construction. We will include a formal discussion of tensor-valued extensions in the revised manuscript.
>
> >Q4: Typo in Figure 4 panel labels.
>
> Thank you for catching this; we will correct it.
>
> [1] Bauerheim, M. Routes towards an effective AI in CFD. Journal of Fluid Mechanics, 2026.

---

> > ### Author Rebuttal · Reviewer_aBYY · 2026-04-03
> >
> > I thank the authors for their clarification. I will raise my score.

---

> > > ### Author Response · Authors · 2026-04-07
> > >
> > > We sincerely thank the reviewer for the increased score and the constructive feedback throughout the discussion. We will incorporate all revisions in the final manuscript, including the explicit symmetry statement in the abstract, the error bound proposition, and the corrected Figure 4 labels.

---

### Official Review · Reviewer_QVCm · 2026-03-12

**Soundness:** 2
**Presentation:** 2
**Significance:** 3
**Originality:** 3
**Overall Recommendation:** 5
**Confidence:** 3

**Summary:**

This paper introduced a new rotationally equivariant method on  Vision Transformer framework for neural PDE solvers called ReViT. Their key idea is to consider local coordinate bases which are "canonical" under special orthogonal transformations, and decode the equivariance after processing, which is realized by three key steps in the archetecture: (1) Local Canonicalization; (2) Invariant Transformer Processing; (3) Equivariant Decoding.

**Compliance With Llm Reviewing Policy:**

Affirmed.

**Final Justification:**

I change my decision from "weak accept" to "accept".

**Key Questions For Authors:**

1) When arguing for the classical representational mismatch, the authors used Schur's Lemma, a very well known foundational lemma in representation theory, but Schur's Lemma is about irreducible representations, but no irreducible representations (or any representation theory of orthogonal groups) were used anywhere in this paper

2)The end of Section 4.4 claimed that "the final output is strictly equivariant", but in Section 4.5 the authors claimed that "our ReViT achieves exact chiral octahedral group O equivariance and approximate SO(3) equivariance", so there's an obvious gap due to the resampling, then is there a theoretical guarantee (like a theorem) about your approximation with nice properties?


3)when using the error metric, was that an optimization problem about choosing one of the 24 elements in O? If so, then there's only an approximate equivariance in the end, but then in what ground is the chiral octahedral group O enough to give a good performance?

**Limitations:**

This paper's method is overall creative, however, the following issues are concerning as well:

(1) Even though there is a promised equivariance in the decoding, the unavoidable grid-based constraints break this equivariance, then the strict SO(3)-equivariance becomes "approximated" O-variance: SO(3) is a Lie group with infinitely many elements, while the octahedral group O is not, so there is a "not small gap" in between; unfortunately, we did not see a sound argument why this reduction is valid.

(2)With this reduction from a Lie group to an approximation with a finite group, a theoretical analysis on this approximation should be given to validate this reduction, but in this paper, there has been no mathematical results (e.g. proposition, theorems, etc.) to give a detailed analysis on this approximation;

(3)The writing is confusing: in Section 4.1 there is a proof of invariance, but there is not a precise mathematical statement before that proof;  when approximating the equivariance, there should be a clear-stated optimization problems with the domain when using the error metric, but it is not even clear from the first reading, nor did the author explain what kind of functions "f" are considered in that optimization problem.

**Strengths And Weaknesses:**

The authors avoided the traditional equivariant approach by studying the behavior of local bases, and their behaviors under orthogonal transformations, while studying the equivariance at the final decoding step, which, in some sense, overcome the challenges of the equivariant methods from the classical ViT frameworks, and the simulation study did show some good results in some situations.

---

> ### Author Rebuttal · Authors · 2026-03-30
>
> We sincerely thank the reviewer for the careful and mathematically rigorous review. We address each concern below.
>
> >Q1: Schur's Lemma is about irreducible representations, but no irreducible representations were used anywhere in this paper.
>
> Yes, Schur's Lemma is invoked in Section 3.3 solely as a *no-go argument*: it explains why standard ViT linear layers cannot be made equivariant by simply constraining them. We do not build an architecture from irreducible representations. We will revise the manuscript to make this clearer.
>
> >Q2: There's an obvious gap due to the resampling, then is there a theoretical guarantee about your approximation?
> L1: The strict SO(3)-equivariance becomes "approximated" O-equivariance [...] we did not see a sound argument why this reduction is valid.
> L2: A theoretical analysis on this approximation should be given [...] there has been no mathematical results.
>
> We address Q2, L1, and L2 together. There are two distinct levels of analysis. **Algebraically**, the pipeline is exactly equivariant: given input $\mathbf{u}$ and $R \in \mathrm{SO}(3)$, the local basis transforms as $\mathbf{B}_i \mapsto R\mathbf{B}_i$, the invariant tokens $\mathbf{z}_i = \phi(\mathbf{B}_i^T \mathbf{u}_i)$ are unchanged, and the decoder reconstructs via $\hat{\mathbf{u}}\_k = \mathbf{B}\_i \hat{\mathbf{u}}\_{\text{local},k}$, yielding $f(R\mathbf{u}) = Rf(\mathbf{u})$. **On a discrete grid**, however, an arbitrary $R \in \mathrm{SO}(3)$ requires interpolation, breaking exact equivariance. This is inherent to *any* grid-based method[1], not specific to ReViT. Based on [3], we provide the following guarantee:
>
> **Proposition.** Let $f$ denote the ReViT mapping, $\mathcal{I}_h$ the grid interpolation operator. For any $R \in \mathrm{SO}(3)$:
> $\|f(\mathcal{I}\_h(R\mathbf{u})) - Rf(\mathbf{u})\| \leq L\_f \cdot \epsilon\_{\text{resamp}}$, where $L\_f$ is the Lipschitz constant [4] and $\epsilon\_{\text{resamp}} = \|\|\mathcal{I}\_h(R\mathbf{u}) - R\mathbf{u}\|\|$ is the interpolation error. By polynomial interpolation theory [2], $\epsilon\_{\text{resamp}} \leq C h^p \|D^{p+1}\mathbf{u}\|$ with grid spacing $h$, interpolation order $p$, and field smoothness $\|D^{p+1}\mathbf{u}\|$. For $R \in O$, $\epsilon\_{\text{resamp}} = 0$ holds. The gap between $O$ and $\mathrm{SO}(3)$ is thus a discretization artifact that vanishes as $h \to 0$.
>
> To the best of our knowledge, ReViT is the first work that achieves exact $O$-equivariance for grid-based neural PDE solvers. Our empirical results on the KF case confirm that ReViT’s approximate equivariance under non‑grid‑aligned rotations remains 26.6% lower (in MSE) than that of augmented baselines. For 3D tasks, ReViT achieves a 96.3% reduction in MSE on the TCF task compared with P3D, demonstrating that hard‑coded equivariance is sufficient for grid‑based neural PDE solvers. We will include this proposition in the revised manuscript.
>
> >Q3: Was the error metric an optimization problem about choosing one of the 24 elements in O?
>
> Thanks for pointing this out. The metric involves no optimization. Following [5], we evaluate the defect independently for each $g \in O$, using $\mathcal{E}(\mathbf{U}; g) = \| f(g \cdot \mathbf{U}) - g \cdot f(\mathbf{U}) \|$ and $R^2(g) = 1 - \frac{\sum_i (f(g \cdot \mathbf{u})_i - g \cdot f(\mathbf{u})_i)^2}{\sum_i (g \cdot f(\mathbf{u})_i - \overline{g \cdot f(\mathbf{u})})^2}$ as a normalized similarity, where $i$ indexes all elements across $N$ test samples, channels, and spatial dimensions. ReViT achieves $R^2=1.0$ across all 24 rotations (Figure 3); baselines exhibit $R^2 \approx 0$ for non-identity ($O_0$) rotations. $O$ is the maximal finite rotation subgroup of $\mathrm{SO}(3)$ compatible with a cubic grid [5].
>
> >L3: The writing is confusing: no precise mathematical statement before the proof in Section 4.1; the error metric domain is unclear.
>
> We will improve this by:
> 1. Adding a formal proposition before the proof in Section 4.1. Let $\phi(\mathbf{B}_i^T \mathbf{U}_i)$ be the patch embedding where $\mathbf{B}_i \in \mathrm{SO}(d)$ is the local canonical basis. Then for all $R \in \mathrm{SO}(d)$: $\phi(\mathbf{B}_i'^T (R\mathbf{U}_i)) = \phi(\mathbf{B}_i^T \mathbf{U}_i)$, where $\mathbf{B}_i' = R\mathbf{B}_i$. This ensures the claim is precisely stated before the proof.
>
> 2. **Sec. 4.5:** Explicitly definig the error metric using the framework from Q3, and that the metric averages over group elements and test samples (i.e no optimization).
>
> [1] Weiler, et al. General E(2)-Equivariant Steerable CNNs. NeurIPS, 2019.
>
> [2] Brenner, et al. The Mathematical Theory of Finite Element Methods. Springer, 2008.
>
> [3] Wang, et al. Approximately Equivariant Networks for Imperfectly Symmetric Dynamics. ICML, 2022.
>
> [4] Fazlyab, et al. Efficient and Accurate Estimation of Lipschitz Constants for Deep Neural Networks. NeurIPS, 2019.
>
> [5] Balla, J. et al. Implicit Augmentation from Distributional Symmetry in Turbulence Super-Resolution. NeurIPS, 2025.

---

> > ### Author Rebuttal · Reviewer_QVCm · 2026-04-04
> >
> > The authors provided very strong explanations in the rebuttal, and we really appreciate their efforts in that. Overall the answers are clear and they did clarify a lot of our initial confusions.
> >
> > The mathematical proposition about the gap between O and SO(3) should be listed in the paper, which is important intermediate step for sure, and the optimization problem which involves every element in group O should be stated clearly to convenience the curious readers like us.
> >
> > Thank you for your patient expositions!

---

> > > ### Author Response · Authors · 2026-04-07
> > >
> > > We sincerely thank the reviewer for the kind words and for acknowledging the clarity of our rebuttal. We are glad that the explanations on Schur's Lemma, the equivariance error bound proposition, and the non optimization based evaluation metric have resolved the initial confusions. As suggested, we will include the formal proposition on the gap between $O$ and $\mathrm{SO}(3)$ and explicitly state the per-element evaluation procedure in the revised manuscript. We thank the reviewer again for the patient and mathematically rigorous feedback that helped us strengthen our paper.

---

### Decision · Program_Chairs · 2026-04-30

**Decision:**

Accept (spotlight)

**Comment:**

A strong paper proposing a rotationally equivariant transformer model for grid-based PDE solvers. The model leverages local bases to achieve equivariance through canonicalization allowing for an efficient standard self-attention which maintains symmetry.  Reviewers praised the clear theoretical derivation and motivation, the design of a new method which includes several novel details each justified by ablation studies. The method shines for its speed and efficiency relative to other equivariant models while also getting strong results across a wide range of 2D and 3D PDEs including turbulence and Magnetohydrodynamics against strong baselines.

Reviewers shared a concern about exact equivariance relative to the octahedral group and only approximate equivariance to SO(3).  The rebuttals clarified that the architecture is theoretically exactly equivariant to continuous rotations with equivariance error arising from signal discretization.